# Nascent craft specialization in the Pre-Pottery Neolithic A? Bead making at Shubayqa 6 (northeast Jordan)

**Mette Bangsborg Thuesen**[1]*, **Hala Alarashi**[2,3☉], **Anthony Ruter**[4☉], **Tobias Richter**[5☉]

**1** Berlin Graduate School of Ancient Studies, Freie Universität Berlin, Berlin, Germany, **2** IMF-CSIC, Barcelona, Spain, **3** Université Côte d'Azur, CNRS, CEPAM, Nice, France, **4** The Globe Institute: Section for GeoGenetics, University of Copenhagen, Copenhagen, Denmark, **5** Centre for the Study of Early Agricultural Societies, University of Copenhagen, Copenhagen, Denmark

☉ These authors contributed equally to this work.
* mettebt93@zedat.fu-berlin.de

**Data Availability Statement:** All relevant data are within the paper and its Supporting Information files.

## Abstract

The emergence of craft specialisation is a key area of interest for archaeologists investigating the socio-economic history and development of past societies. In southwest Asia, as elsewhere, the origins of craft specialisation have been associated with the emergence of surplus food production, households and social stratification. We present evidence for nascent skilled production of green stone beads at the Pre-Pottery Neolithic A (PPNA) site Shubayqa 6, northeast Jordan. Thousands of pieces of debitage, roughouts and finished beads exhibit signs of standardised production that was probably geared towards exchange. This hints towards incipient skilled craft production that was likely part-time and seasonal. We therefore argue that the appearance of specialist artisans in this autonomous and non-hierarchical society has no correlation with surplus food production, households, or social stratification.

## Introduction

The earliest appearance of craft specialists is commonly associated with the emergence of sedentary, surplus food-producing and possibly hierarchical, socially complex societies. Traditional models have emphasised that surplus food production was a necessity for the emergence of 'serious' specialists, in the sense of individuals or groups that spend a majority of their time producing standardized objects of high quality for consumption by people other than themselves [1, 2]. While this criterion has remained influential in the archaeological literature, today there is a consensus that craft specialisation needs to be understood as a far more complex phenomenon. It is therefore studied as a multivariable continuum that is measurable through group affiliation (independent vs. attached specialists; see Brumfield & Earle [3]; Clark & Parry [4]; Stein [5]), time dedication (part or full-time specialists, see Kerner [6]), the organization of the production (domestic, in workshop, at local or regional levels, see Costin [7] and the quantity or products per specialist and their qualities measured by the degrees of

**Funding:** The research reported in this paper was enabled by grants from the Independent Research Fund Denmark (DFF – 4001-00068 and DFF-8018-00133B), the Danish Institute in Damascus and the H.P. Hjerl Mindefondet for Dansk Palæstinaforskning. URL: https://dff.dk/en http://damaskus.dk/ The funders had no role in study design, data collection and analysis, decision to publish, or preparation of the manuscript.

**Competing interests:** The authors have declared that no competing interests exist.

sophistication and standardization [7]. Craft specialisation is furthermore no longer seen as a purely economic activity, but also as a mode of production entangled with social dimensions. In the recent literature more attention has therefore been given to identifying the social roles of the craft specialists and the use of craft products as an active agent in the shaping of symbolic codes shared by different socio-political actors [8–11].

As in other parts of the world, archaeologists have generally associated the emergence of craft specialisation in southwest Asia with the rise of socially complex societies. However, several studies have shown that this occurs in various forms throughout the prehistory of this region, stressing that this phenomenon is not restricted to later periods associated with urbanisation. Kerner [6], for example, situated the emergence of craft specialisation in the specialist household production of ceramics during the Early Chalcolithic of the Levant, moving away from a previously dominant focus on metal production [12]. Some have argued that there is already clear evidence for craft specialisation during the PPNB [13], ranging from flint knapping [14] and basketry [15] to plaster production[16]. The appearance of bidirectional naviform blade technology that used non-local, high-quality flint for the production of blades during the Early PPNB has been associated with the potential arrival of traveling craft specialists from the northern Levant [14], where it emerged during the late PPNA [17]. Kirkbride [18] also alluded to the possibility of craft specialisation during the LPPNB, when she described the corridor houses from Level II and III at Beidha as workshops for specialised bone and bead manufacture. Another domain of clear signs of craft specialisation during the PPNB is the production of beads [19–21].

The earliest potential evidence for the use of beads in the Levant are naturally perforated *Glycymeris* bivalves from Qafzeh cave dated to around 120,000 cal BP [22, 23]. During the Upper- and Epi-Palaeolithic, beads were made from marine and freshwater shells, and animal bones and teeth [24–27]. The Natufian (ca. 14,500–11,500 cal BP) represents a threshold in the evolution of body ornaments. In addition to little modified shells, bones and teeth, new types appear with geometric or imitation of natural shapes [28–32]. Although the use of stones is attested during this timeframe [33, 34], it is during the subsequent PPNA (ca. 11,750–10,500 cal BP) that stone becomes a dominant raw material for bead production and an important component of the ornamental traditions in Northern Mesopotamia and the Levant [34–36].

In the PPNB, another important threshold was crossed, when semi-precious stones were introduced in bead crafting. Stone bead making flourished thanks to unprecedented technological skills that allowed for the transformation of all kinds of stones including very hard, resistant, and delicate varieties into a wide range of bead shapes and types [19, 37, 38]. This is for example, evidenced by Groman-Yaroslavski and Bar-Yosef Mayer's use-wear analysis of two Middle PPNB stone beads from Nahal Hemar cave that revealed the use of lapidary technology, which made them argue that "*The fully developed manufacturing process reflects a well-established specialized craft*" [39]. At el-Hemmeh, Raad and Makarewicz [40] also recognised technological strategies in bead manufacturing towards more efficiency in their analysis of the Late PPNA and Late PPNB beads.

Wright et al.'s study [11, 41] of the Late PPNB/PPNC bead production workshop at Wadi Jilat 13 and 25 led them to suggest that the emergence of nuclear households, fully-fledged food-producing economies, and increasing social complexity, resulted in the appearance of specialised bead makers. They argued that beads became an important means to construct and assert social identities towards the end of the PPNB, which fuelled and resulted in the emergence of specialist production centres as well as household specialisation. Baysal [42] argued that although the evidence for specialised production of beads at Bonçuklu Höyük in central Anatolia was not conclusive that we must not assume that craft specialists necessarily resembled fully established artisans during the incipient stages of specialization. While skilled

individuals may have spent a great deal of their time producing one type of item intended for exchange or trade, it may be difficult to formally recognize the activities of such individuals archaeologically. More importantly, Baysal [43] highlighted that we should not make *a priori* assumptions under which conditions incipient craft specialists may have become established in the human past, but to focus on the detailed analysis of assemblages and technologies to determine degrees of standardisation, knowledge, surplus production, exchange and the attributed value and meaning of products.

In this study, we present evidence for the skilled manufacture of beads at Shubayqa 6 in Jordan, dated to between ca. 12,400–10,600 cal BP, thereby making it the earliest lapidary bead production site in the Levant known so far. The occupation spans the transition from the Late/final Epipalaeolithic to the end of the Pre-Pottery Neolithic A (PPNA) and the early Pre-Pottery Neolithic B (EPPNB), a period during which the transition from gathering and hunting to plant cultivation occurred in southwest Asia. During this transition surplus food production was not assured or a given [44]. Although there is evidence for food storage facilities at some sites in southwest Asia during this period [45, 46], how significant food surpluses were before the cultivation of domesticated plants is unclear. Despite the recovery of some sickle blades [47] and copious amounts of ground stone tools from Shubayqa 6 [48], no storage facilities or clear archaeobotanical evidence for plant cultivation have been identified to date.

Our study provides an unprecedented amount of early evidence for bead manufacturing and suggests that the stone bead production at Shubayqa 6 displays signs of skilled, standardized production that is indicative of nascent craft specialisation. These results highlight how such a production appeared considerably earlier than previously thought, and how it was not necessarily connected to surplus food production. We therefore argue that alternative scenarios for the emergence of craft specialisation in southwest Asia should be considered.

## Shubayqa

Shubayqa 6 is located in the volcanic lava field of the *Harrat al-Sham* (Black Desert) in Northeast Jordan, ca. 130 km east of Amman (see Figs 1 and 2). The settlement is situated at the northern edge of the Qa' Shubayqa mudflat from which the site takes its name. The area has produced evidence for a dense concentration of late Pleistocene and early Holocene sites dating from the Early Natufian (c. 14,500–14,000 cal BP) to the end of the PPNA (c. 11,600–10,500 cal BP.). The site of Shubayqa 6 forms a settlement mound measuring ca. 3.000 $m^2$ in an area that rises ca. 1.5–2 m above the surrounding area. Excavations have to date focused on a 10 x 10 m large area on the east side of the mound. Radiocarbon dates, chipped stone artefacts and stratigraphy currently suggest occupation stretching from the end of the Epipalaeolithic to the end of the PPNA and the EPPNB. The site has three major occupation phases, distinguished on the basis of the site's stratigraphy, lithic typology and radiocarbon dates. The earliest occupation is a final Epipalaeolithic/Khiamian phase, characterised by the presence of both short lunates, el-Khiam points and many drills. The middle phase dates to the early PPNA and is characterised by fewer lunates (compared to the preceding phase), some el-Khiam points and many drills. The last and uppermost phase dates to the late PPNA/early PPNB, with a chipped stone assemblage characterised by a greater variety of points—including Helwan points—with lunates being virtually absent. The earliest phase, final Epipalaeolithic/Khiamina, was radiocarbon dated to 12,300–11,800 cal BP (all dates reported here are the summed calibrated dates for each phase at 68.2% confidence, calibrated in OxCal 4.4. [49] using IntCal20 [50]. The early PPNA phase is dated to ca. 11,900–11,200 cal BP, while the late PPNA/early PPNB phase dates to 11,090–10,600 cal

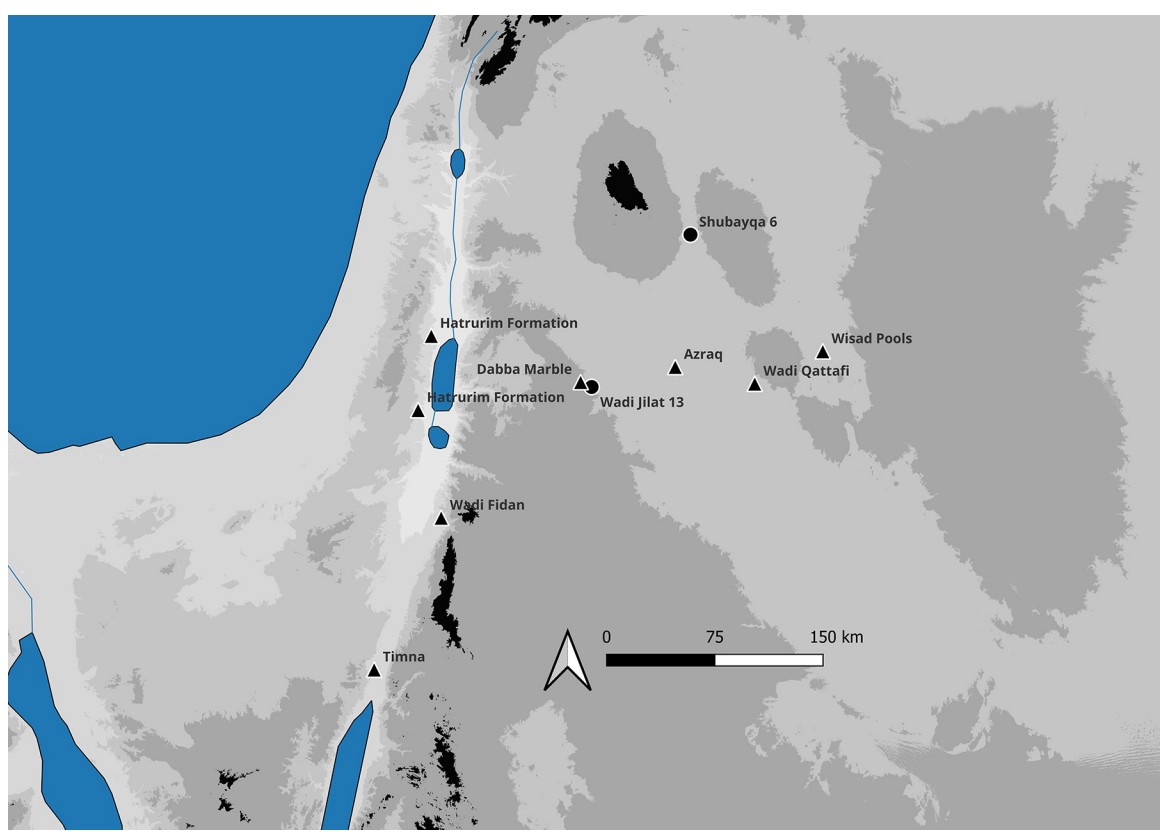

**Fig 1. Map of the southern Levant showing the location of Shubayqa 6 and other sites mentioned in the text, including greenstone raw material sources (marked by triangle).** Phase plans produced by Lisa Yeomans (amended from Yeomans et al. [52], Fig 2 and Yeomans et al. [53], Fig 1; additional radiocarbon dates placing Space 7 in the earlier architectural phase). Created in QGIS version 3.32.0 using Shuttle Radar Topography Mission (SRTM) data (doi:/10.5066/F7F76B1X) and vector datasets from Natural Earth (https://www.naturalearthdata.com/).

BP. The Khiamian and EPPNA phase was uncovered in the main excavation area and is associated with two larger oval, stone-built structures (Space 1 and Space 4) with stone paved floors, as well as some smaller adjacent structures. The LPPNA/EPPNB phase is associated with two fully excavated smaller, oval buildings with trampled floors (Space 3 and Space 10), and several other partially excavated buildings. Finds include chipped stone tools and waste from chipped stone manufacture, numerous ground stone artefacts, worked bone and faunal remains, as well as botanical remains. Current evidence suggests that the site was intensively occupied and that people spent extended periods at the site. Although one should be cautious with the application of specific terms such as 'village' [51: p.79-81], Shubayqa 6 should probably not be understood as a transitory camp, but as a place in which a community invested considerable time and resources in. Excavations were carried out using the single-context recording method. Deposits and layers spread over a larger area, e.g. middens, deposits above floors, were excavated in a 1 x 1 m grid system using arbitrary excavation spits. All excavated sediments were dry-sieved on site using 0.5 x 0.5 cm mesh. The dry-sieve residue was collected, washed, and sorted in the field camp. In addition, bulk sediment samples were collected for archaeobotanical flotation, and all 'heavy residue' collected in the flotation tank was washed and sorted in the field station. Thus, we are certain of a high recovery rate of even small beads and bead fragments.

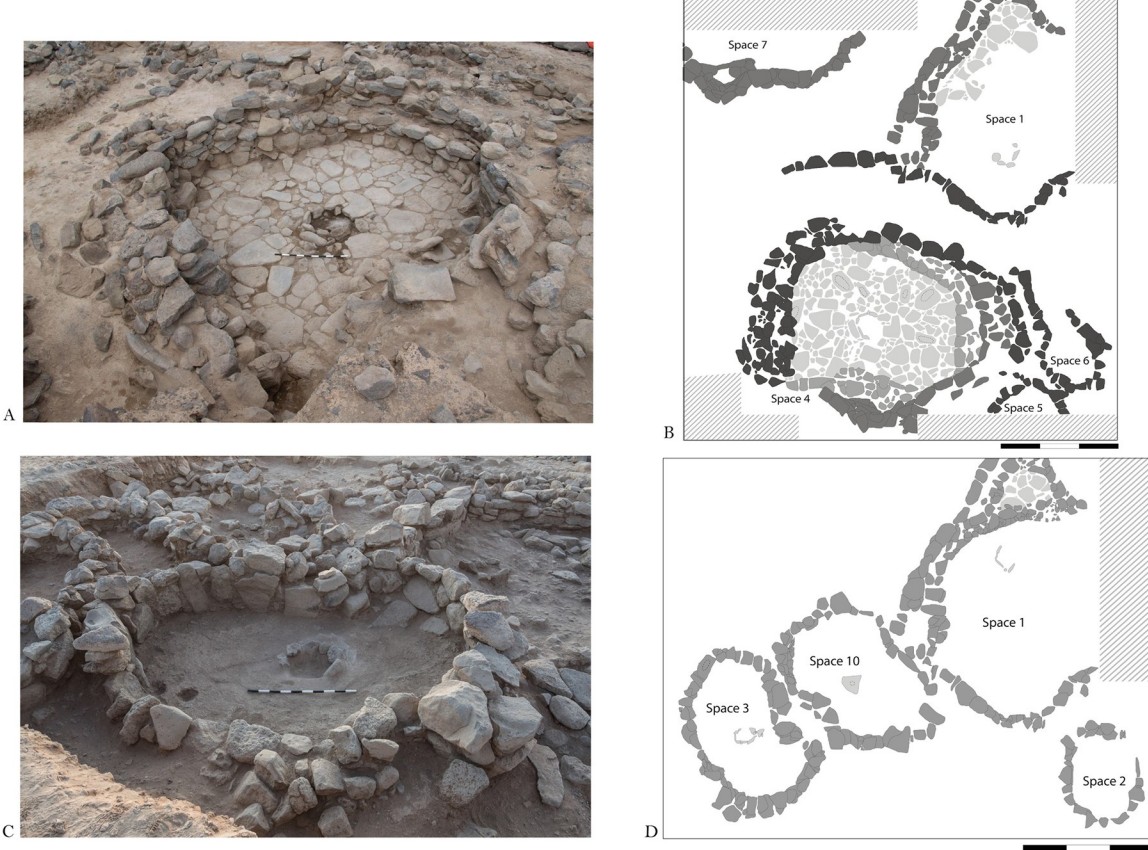

**Fig 2. Plan of the excavation at Shubayqa 6.**

## Materials and method

The studied assemblage comprises nearly 2500 beads and bead roughouts from Shubayqa 6. In addition to finished beads and roughouts, the dry sieving of all excavated sediment, as well as extensive sampling of sediment for flotation, resulted in the recovery of large amounts of greenstone debitage and raw material nodules. This material has thus far not been fully quantified, but initial analysis of a selected group revealed more than 13,695 fragments of production waste with a substantial amount still remaining to be counted. Most of the material is from midden deposits that accumulated inside Space 1 and 4. Finished beads, roughouts, debitage and nodules were recovered from all excavated contexts, although there are clear concentrations associated with Spaces 1 and 4 (EPPNA), and Space 3 (LPPNA). Spaces 2, 5, 6, and 7 contained relatively few beads and these mostly included finished disc beads and disc bead roughouts.

In this study, the analytical method is based on the *châine opératoire* approach [54, 55] implemented on beads (e.g. Alarashi [20]; Barthélémy de Saizieu and Bouquillon [56], where we examine the diversity of raw material used and their potential sources, the production sequences, and–to a lesser extent–a general assessment of use-wear of the finished stone beads. The analysis of the production sequence relied on recording of quantitative and qualitative data. The latter was based on a classificatory approach that categorised artefacts into different production stages. Unfinished beads were registered as roughouts while finished beads were further sub-divided into distinct typological categories following the classification system

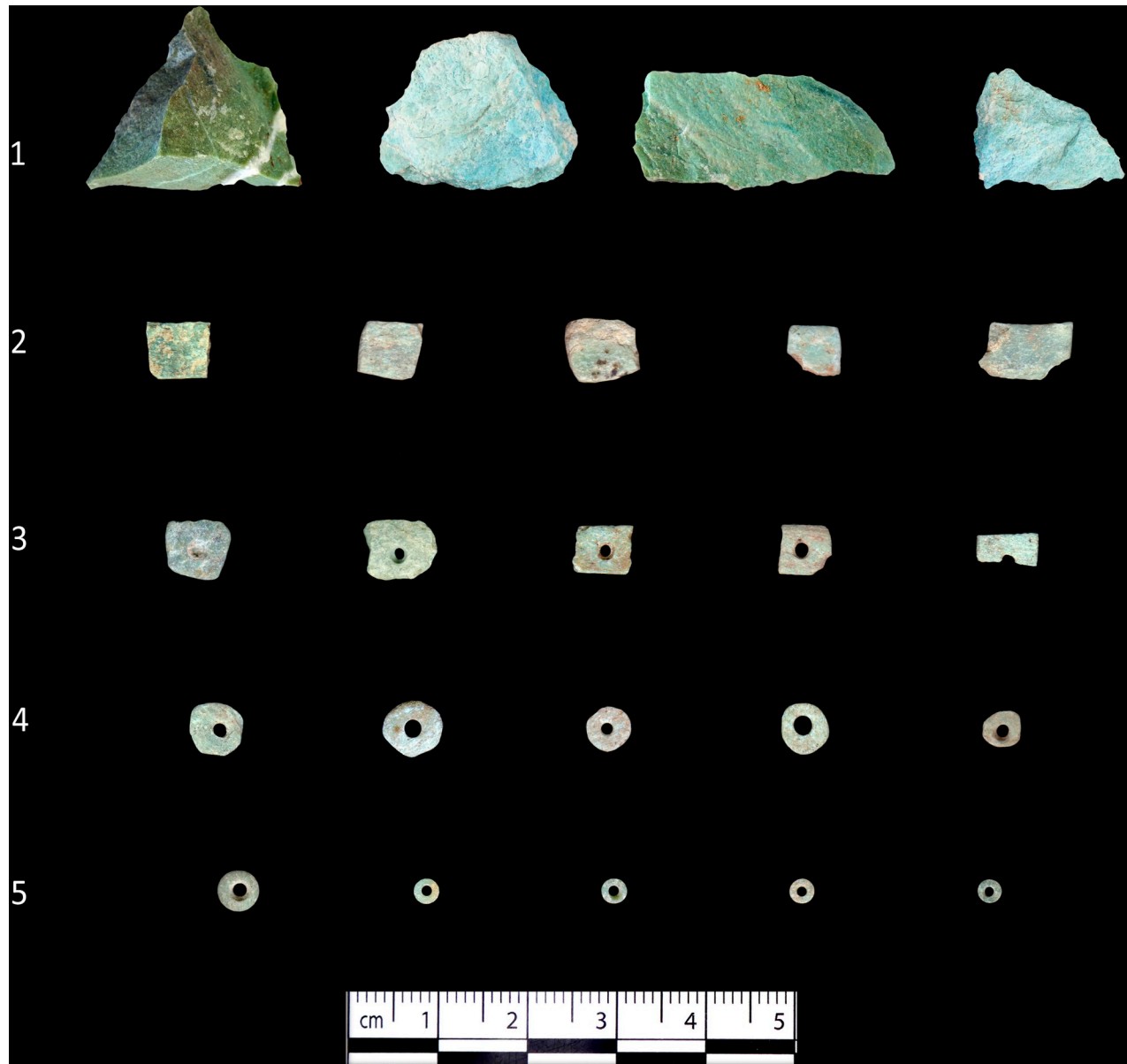

**Fig 3. Greenstone roughouts and beads recovered from Shubayqa 6 divided according to the five identified production stages.**

developed by Wright & Garrard [11] supplemented in some cases by additional categories drawn from Beck [57], Bains [58], and Bar-Yosef Mayer [33]. The vast majority is defined as disc beads in both finished and unfinished forms.

Among the typological diversity, only disc beads occur in different stages before the final products. This allowed the reconstruction of their production sequences and the identification of their order (Fig 3). Partially inspired from previous works [41], a total of five manufacturing stages were established:

1. The extraction of fragments of raw materials (nodules);

2. The shaping of tabular roughout (tabulars);

3. The drilling of the tabulars (drilled tabulars);

4. The shaping of pseudo-circular shapes of the drilled tabulars (unfinished disc beads);

5. The finishing stage of the disc beads (final products).

The macroscopic observations were supplemented by microscopic analysis to record more minute production traces, such as drilling and abrasion patterns. The variables for the microscopic analysis were defined according to the same system used by Bains [58] for her studies on the stone beads from Çatalhöyük that include drilling characteristics, perforation morphology, characterisation of perforation marks, as well as marks on the length/height (the profile of the bead) or the ends (perforated surfaces) of the beads, shape of edges, and evidence of use. A total of 110 stone beads were sampled for this part of the analysis, conducted using a Brunel SP-400 incident light microscope and a GXM MZSTLED stereo microscope. The specimens were selected based on stratigraphic context to ensure that a representative sample for each occupation phase and production stage was studied.

These observations were supplemented by a small experimental program for which the production of greenstone beads was replicated during the 2019 Shubayqa field season to further test different methods and techniques (S2 Appendix).

Because the procurement of raw materials is a significant component of the labor invested in production, we sought to describe the variance in raw materials used for beads to infer something about their raw material and potential sources. To this end, we selected 33 beads from the Shubayqa 6 assemblage for an amended (non-destructive) petrographic analysis, utilizing stereomicroscopy to describe mineralogy and texture and a NITON XL3 GOLDD + pXRF energy dispersive x-ray analyser to estimate elemental composition. Three additional greenstone samples from the Wadi Jilat, a known prehistoric quarry were also analysed to make a preliminary comparison with the apatitic limestone specimens from Shubayqa. The composition in parts per million (PPM) of 40 elements plus balance (Bal), the amount of the signal that the instrument is unable to attribute to an element (roughly proportional to the sample mass of the unmeasured light elements), was generated for each of the 36 samples plus two modern ostrich eggshell controls. All measurements were made at standard room temperature and pressure (approx. 1 bar) without helium purge or vacuum, using a test stand. The specimens were cleaned with deionized water and measured directly without sample pots. The resulting data were processed with "Standard Thermo Scientific™ Niton Data Transfer (NDT™) PC software suite" set for "TestAll Geo" at four separate runs of: light elements, medium heavy elements, heavy elements and standard elements each measured for 120 seconds. Certified standard reference samples including CCRMP TILL-4PP (180–646) and NIST 2709a (180–649) and a single piece of obsidian from the Fantale stratovolcano in Ethiopia, previously analyzed with ICP-MS were used to gauge the accuracy of the elements measured in this analysis. None of the measurements had greater than a 5% error compared to these standards.

The principal objective of this part of the study was to assess the diversity of the assemblage by gauging how many different types of materials were used in the bead production at Shubayqa 6 and to measure the degree of homogeneity in the 'greenstone' category. This limited exploratory characterization is also an initial step towards the identification of raw-material sources, but that is beyond the scope of this study.

## Results

We recorded 2398 beads and bead roughouts from Shubayqa 6 as part of this study. Of the finished and unfinished stone beads, which we focus on here, 533 were recovered from early

**Table 1. Heat map correlating the bead typology with percentage distribution across the occupational phases.** Actual numbers are marked in brackets.

| Bead type | EPPNA | LPPNA | Post-Neolithic/Mixed |
|---|---|---|---|
| Disc beads | 42,1% (197) | 36,61% (157) | 32,26% (342) |
| Oval beads | 0% | 0% | 0,08% (1) |
| Tubular beads | 3,81% (20) | 6,75%% (33) | 2,13% (27) |
| Cylinder beads | 0,38% (2) | 0,2% (1) | 0,16% (2) |
| Barrel beads | 1,52% (8) | 0,2% (1) | 0,71% (9) |
| Pendants | 0,19% (1) | 0,82% (4) | 0,16% (2) |
| Double perforated beads | 0,19% (1) | 0,2% (1) | 0,47% (6) |
| Toggle beads | 0% | 0% | 0,08% (1) |
| Perforated pebbles | 0% | 0% | 0,16% (2) |
| Indeterminate | 0,57% (3) | 0,2% (1) | 0,55% (7) |
| Roughouts[a] | 51,24% (269) | 55,01% (269) | 63,25% (802) |

[a]roughouts represent preforms of disc beads.

PPNA (EPPNA) contexts, 501 from late PPNA (LPPNA) contexts. 1307 came from topsoil or disturbed contexts and are thus less secure. Since this bead-making technology is quite specific to the Pre-Pottery Neolithic, and since there was no occupation at the site between the start of the EPPNB and the Early Bronze Age we feel confident that this material is broadly related to the PPNA occupation (see **Table 1**). Roughouts make up the largest single group accounting for 50.4% in the EPPNA and 53.7% in the LPPNA phases. The most common type are disc beads, which account for 36.6% and 42.1% of the beads in the EPPNA and LPPNA phases respectively.

## The finished beads

Among the studied assemblage 931 beads were assigned to a specific type, while the remainder were classified as roughouts (n = 1340). The most common types are shown in Fig 4. The majority of beads (n = 809) belong to the disc beads that are characterised as flat beads of circular shape. Although these were predominantly made from stone, other kinds of raw material, such as ostrich eggshell, bone and clay, were also used. Other common bead stone types include barrel beads, cylinder beads (cylindrically shaped with narrow perforation), double perforated beads (flat beads of rectangular shape), tubular beads (cylindrically shaped with wide perforation), and pendants (triangular or trapezoidal shaped beads) although they are comparatively rare and all made from greenstone. Finally, oval beads (flat beads of oval shape), toggle beads (dumbbell shaped) and perforated pebble were each represented with just 1–2 specimens.

To assess the degree of standardization in the Shubayqa 6 disc bead production, we measured overall bead diameter and maximum thickness of all complete beads from secure EPPNA and LPPNA contexts (N = 96). The result (**Fig 5**) indicates that there is no significant size difference for EPPNA or LPPNA disc beads. The smallest bead measures 3 mm in diameter with a minimum thickness of 0.8 mm. There is a clear clustering between 3–6 mm diameter and 0.5–1.5 mm in thickness. Beads with greater diameter and of greater thickness exist from both phases, but more so from the EPPNA, while LPPNA beads cluster more tightly. However, this does not necessarily suggest greater standardization, as overall sample sizes are too low to make this inference. Nevertheless, the clustering of EPPNA and LPPNA beads does suggest that bead makers worked towards a specific size template of small and thin beads. In this

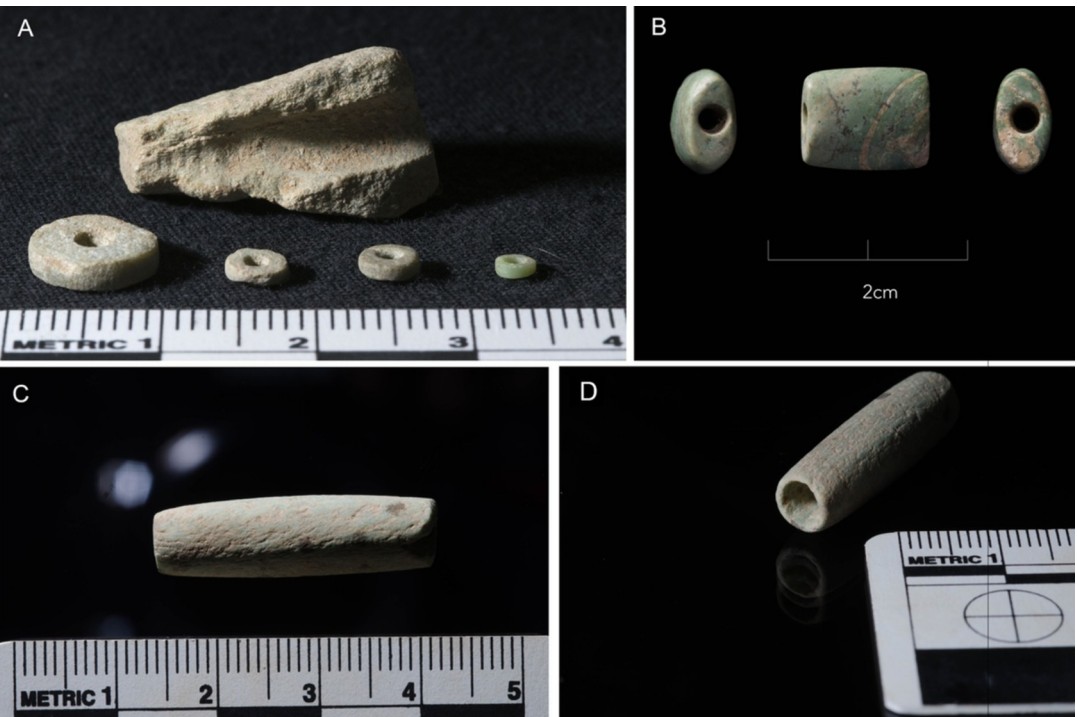

**Fig 4. Most common types of stone beads found among the Shubayqa 6 assemblage.** (A) Side view of disc beads of various sizes with broken roughout (B) barrel bead with profile and both ends shown (C) profile of tubular bead (D) one end of tubular bead.

sense, the Shubayqa 6 bead production can be seen as standardised during both the EPPNA and LPPNA.

The majority of beads examined under the microscope did not show traces of use. Only five finished beads displayed wear marks, including widened holes in the perforated areas and unevenly smoothened surfaces on the ends of the beads, which could derive from the beads being worn. Therefore, there is at present little evidence of use-wear, but given that the sample size was small, future work may provide more results.

## Raw materials

Of the 2398 beads recovered, 1897 (79.1%) were made from rocks and minerals, while the remaining 20.9% were produced from hard animal tissues, namely ostrich eggshell, marine shells and animal bones (see Fig 6). Among the first group, 72.8% were classified as "greenstone", a category that in the Levant includes different types of greenish rock and minerals such as apatitic limestone varieties, malachite, turquoise, amazonite and copper silicates [59]. Other 6% of the total number of the beads from this group could not be classified yet, while less than 0.5% were identified as basalt, flint, clay, and ochre.

This basic classification is augmented by the XRF analysis of selected specimens from Shubayqa and geological samples from the Wadi Jilat. Fig 7 lists PPM values from each sample for eight of the 40 elements measured in the XRF analysis. These elements were selected for the table because they are major constituents and differ significantly between the material classes. The PPM values of balance (Bal) are also listed. High concentration of both calcium (Ca) and phosphorus (P) in most of the greenstone specimens as well as the samples from Wadi Jilat,

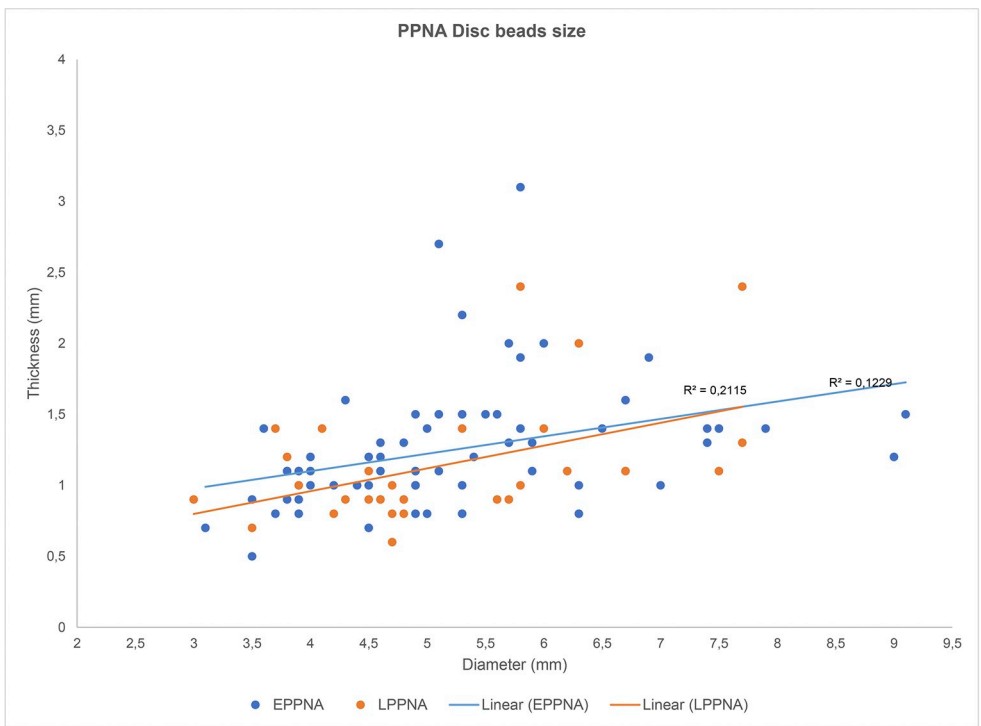

**Fig 5. Scatter plot of the PPNA disc beads from Shubayqa 6.** The general size is represented by the relation of the diameter and the thickness. Lineal regressions of the EPPNA and LPPNA beads show similar distributions with a slight smaller size of the LPPNA.

confirm that these are apatitic limestone, consistent with the green Dabba Marble identified in other Neolithic contexts in the southern Levant [11, 41]. In contrast, calcite and eggshell specimens are distinguished by high concentrations of calcium alone. High concentrations of copper (Cu), as well as manganese (Mn), distinguish the copper minerals, while the mass of the siliciclastic specimens are largely comprised of silica (Si) and other clay minerals including aluminium (Al) and potassium (K). The PPM estimates for 40 elements plus balance generated for each sample were used to generate the Principle Component analyses presented in (Fig 8A and 8B).

Fig 8A plots the Principal component analysis (PCA) scores of all 36 samples on the first and second components calculated on the variance-covariance matrix in which the influence of each element is equivalent to its compositional proportion. This is useful for sorting very different classes of material (e.g. limestone and metal ores and siliceous rock).

The apatitic limestone specimens generally plot from the center to the upper right quadrant due to their concentrations of both calcium and phosphorus. A second adjacent cluster in the lower right is formed from samples that share the high concentration of calcium but have relatively less phosphor, consistent with their identification as precipitated carbonate rock (probably calcite) and ostrich eggshell. A third cluster is distinguished by relatively high concentrations of silicon (Si), aluminium (Al), potassium (K) and iron (Fe) and very little calcium (Ca), consistent with their petrographic designations of siliciclastic rock. These including sandstones, silicified mudstone or slate, and hard crypto-crystalline silicate rock, which could occur as a precipitate within a limestone facies.

Two of the 36 specimens (Shub-2a and Shub-23) analysed with XRF were distinguished by relatively high copper content (587,095 and 327,396 ppm Cu respectively), which is well within

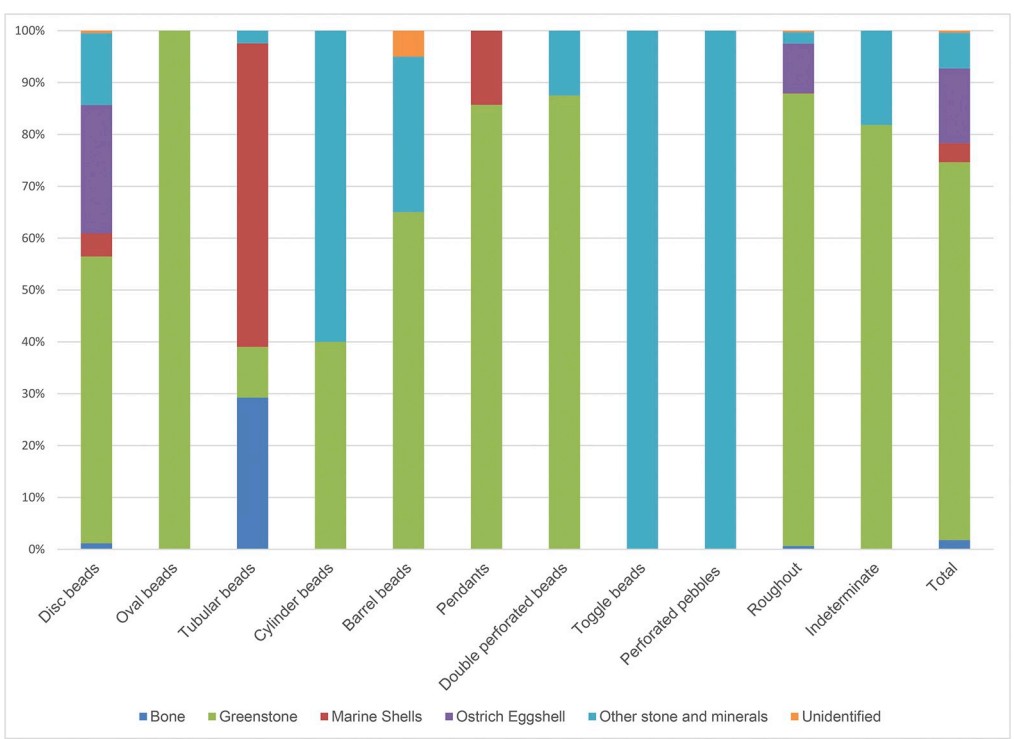

**Fig 6. Distribution of raw material according to the bead typology.**

the range of commercial copper ores. This coupled with the relatively low concentrations of sulphur (S), iron (Fe) and phosphor (P), make malachite, which is a copper carbonate hydroxide mineral, the most likely candidate for both specimens. However, chrysocolla cannot be ruled out for the second sample (Shub-23) because both silicon (Si) and aluminum (Al) are significant constituents. Specimen Shub-2a has a significantly higher copper content (587,095 ppm), yet no trace of barium or lead consistent with relatively pure malachite, which does outcrop in the Wadi Faynan and Wadi Fidan area. The significant variance in these trace elements might be used to match the two copper-bearing specimens to their respective ore sources, when a reference database is available.

Fig 8B compares the composition of Shubayqa specimens designated as greenstones along with three apatitic limestone samples from the Wadi Jilat by plotting the sample scores of 40 elements measured with XRF of the first and second components, which explain 32.8% and 21.3% of the total variance respectively. The PCA is calculated from the correlation matrix, which assigns equal weight to all elements, including those with "trace" concentrations. This is useful when comparing archaeological material to geological samples from potential sources. As noted above, the copper minerals are again clearly distinguished by their relatively high concentration of copper as well as potassium (K), aluminium (Al), titanium (T), lead (Pb), iron (Fe), manganese (Mn) and magnesium (Mg) relative to calcium (Ca) and phosphorus (P) comprising the limestone specimens. Variability in the Shubayqa limestone is driven by a suite of elements including antimony (Sb), tin (Sn), chromium (Cr) and strontium (Sr). Samples from Wadi Jilat are homogeneous and cluster close to a subset of Shubayqa samples relatively enriched in these elements along with unmeasured light elements. This cluster has less of the trace elements tungsten (W) and molybdenum (Mo) as well as the principle constituents of apatitic limestone, calcium and phosphorus. Although we have far too few specimens to

| Site/Sample | Classification | SELECTED ELEMENTS PPM Si | Al | K | P | Ca | Mn | Fe | Cu | Bal |
|---|---|---|---|---|---|---|---|---|---|---|
| SHUBAYQA | | | | | | | | | | |
| Shub-2a | Greenstone (Copper mineral) | 17548 | 4448 | 405 | 119 | 66139 | 4431 | 188 | 587096 | 315563 |
| Shub-23 | Greenstone (Copper mineral) | 128837 | 55038 | 18107 | 695 | 9486 | 8378 | 10855 | 327396 | 381230 |
| Shub-2b | Greenstone (apatitic limestone) | 54574 | 16932 | 4174 | 68026 | 385294 | 234 | 3444 | 212 | 457787 |
| Shub-4.1 | Greenstone (apatitic limestone) | 24616 | 6247 | 1345 | 95255 | 436822 | 1 | 1310 | 39 | 428624 |
| Shub-4.2 | Greenstone (apatitic limestone) | 24961 | 7110 | 1118 | 91759 | 445972 | 1 | 1556 | 94 | 417135 |
| Shub-4.3 | Greenstone (apatitic limestone) | 12839 | 2068 | 917 | 135360 | 409760 | 1 | 672 | 32 | 435435 |
| Shub-4.4 | Greenstone (apatitic limestone) | 16088 | 2954 | 644 | 90585 | 451804 | 1 | 1250 | 67 | 432571 |
| Shub-4.5 | Greenstone (apatitic limestone) | 18462 | 2313 | 806 | 82532 | 555262 | 1 | 595 | 124 | 332979 |
| Shub-4.6 | Greenstone (apatitic limestone) | 36673 | 9007 | 1448 | 90474 | 394345 | 1 | 2208 | 84 | 455207 |
| Shub-4.7 | Greenstone (apatitic limestone) | 22582 | 3365 | 1098 | 99169 | 466135 | 1 | 1066 | 62 | 385824 |
| Shub-19.1 | Greenstone (apatitic limestone) | 14472 | 3082 | 698 | 156800 | 431094 | 302 | 735 | 72 | 387789 |
| Shub-19.2 | Greenstone (apatitic limestone) | 12240 | 2309 | 570 | 149463 | 431950 | 224 | 876 | 42 | 390731 |
| Shub-19.3 | Greenstone (apatitic limestone) | 12393 | 1550 | 572 | 99796 | 437964 | 67 | 335 | 13 | 442639 |
| Shub-21 | Greenstone (apatitic limestone) | 31840 | 8718 | 2065 | 55309 | 386408 | 218 | 1704 | 31 | 504937 |
| Shub-20.2 | Greenstone (apatitic limestone) | 33561 | 9585 | 986 | 105357 | 474531 | 1 | 2635 | 129 | 363721 |
| Shub-20.1 | Greenstone (apatitic limestone) | 17853 | 1 | 726 | 102996 | 455476 | 1 | 422 | 98 | 416219 |
| Shub-22.1 | Greenstone (apatitic limestone) | 38126 | 14260 | 3207 | 119395 | 432855 | 1 | 2300 | 52 | 384276 |
| Shub-22.2 | Greenstone (apatitic limestone) | 24851 | 8813 | 1645 | 30433 | 455453 | 272 | 1752 | 83 | 470016 |
| Shub-11 | Calcite | 22593 | 5092 | 1039 | 376 | 438458 | 1 | 2062 | 1 | 527700 |
| Shub-12 | Calcite | 20362 | 5435 | 1086 | 3162 | 448381 | 813 | 3342 | 83 | 514928 |
| Shub-14 | Calcite or shell | 26781 | 6626 | 4034 | 8270 | 390977 | 510 | 5704 | 58 | 549882 |
| Shub-15 | Calcite or shell | 18802 | 5389 | 1275 | 2144 | 446721 | 502 | 1041 | 1 | 521426 |
| Shub-17 | Ostrich egg shell (Shubayqa) | 21241 | 6720 | 1335 | 1 | 462986 | 1 | 1110 | 1 | 505238 |
| Shub-18 | Ostrich egg shell (Shubayqa) | 42681 | 12845 | 3089 | 1 | 434020 | 1 | 3476 | 24 | 500889 |
| Shub-5 | Ostrich egg shell (Shubayqa) | 28381 | 8339 | 2694 | 757 | 442178 | 107 | 1982 | 24 | 512081 |
| control 1 | Ostrich egg shell (zoomus) | 2125 | 1 | 403 | 2068 | 453664 | 69 | 81 | 8 | 534834 |
| control 2 | Ostrich egg shell (zoomus) | 3491 | 2344 | 1915 | 1970 | 454780 | 90 | 119 | 17 | 517900 |
| Shub-1 | Siliciclastic rock | 295417 | 9164 | 7291 | 8275 | 29034 | 1 | 1789 | 10 | 639408 |
| Shub-6 | Siliciclastic rock | 220214 | 38328 | 25973 | 1 | 13700 | 236 | 61949 | 290 | 627151 |
| Shub-7 | Siliciclastic rock | 207271 | 31573 | 16458 | 794 | 2958 | 283 | 46687 | 94 | 670965 |
| Shub-13 | Siliciclastic rock | 198488 | 48734 | 37532 | 1267 | 21442 | 1129 | 78226 | 200 | 591185 |
| Shub-8 | Mafic? | 108198 | 56907 | 1458 | 865 | 7167 | 3071 | 160050 | 1 | 621933 |
| Shub-9 | Unclassified | 48711 | 63628 | 3239 | 90991 | 90547 | 1 | 3582 | 71 | 688663 |
| Shub-10 | Unclassified | 92221 | 27486 | 12801 | 21805 | 156488 | 333 | 13524 | 56 | 658625 |
| Shub-16 | Unclassified | 62416 | 7705 | 6600 | 574 | 260952 | 393 | 7254 | 41 | 645050 |
| WADI GILAT | | | | | | | | | | |
| WG1 ave | Greenstone (apatitic limestone) | 15966 | 3932 | 703 | 52167 | 338636 | 122 | 904 | 30 | 579349 |
| WG2 ave | Greenstone (apatitic limestone) | 11689 | 3340 | 368 | 49589 | 357286 | 286 | 838 | 50 | 565090 |
| WG3 ave | Greenstone (apatitic limestone) | 12176 | 3266 | 379 | 54673 | 350038 | 95 | 747 | 24 | 568499 |

**Fig 7. Lists PPM values from each sample for eight of the 40 elements measured in the XRF analysis.**


adequately describe the variability of the Wadi Jilat source, this preliminary analysis suggests that while some of the Shubayqa limestone may have originated there that was not the case for the majority of samples. The limestone at Shubayqa is variable and was probably sourced from several different quarries.

In summary, The XRF analysis showed that most of the greenstone at Shubayqa 6 was Dabba Marble, although other types of greenstone are also present in the assemblage. The analysis also identified copper ores, which do not occur in the Azraq Basin or Harra. Other minerals used for bead making appear to be travertine, sandstone and silicified mudstones. Most of the stone samples that were studied for XRF did not measure more than 3.5–4 on Mohs hardness scale and are therefore considered soft stones that are easy to work. Thus, artisans clearly preferred soft stones with greenish or reddish colour.

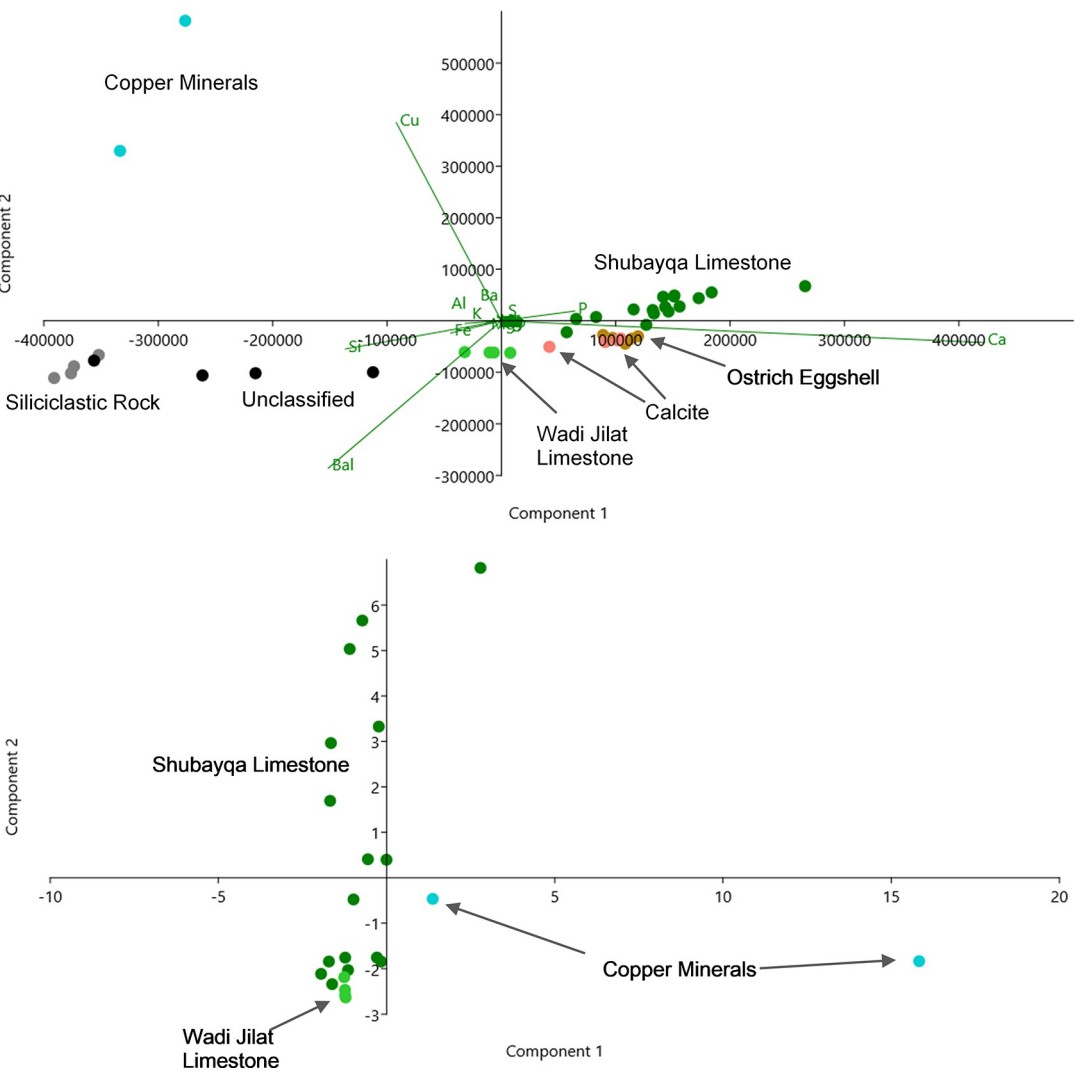

**Fig 8. A.** The PCA biplot generated from the XRF analysis of all Shubayqa specimens and several from Wadi Jilat along with two modern ostrich eggshell controls. **B.** The PCA biplot calculated on the correlation matrix generated from the XRF analysis of the Shubayqa specimens designated as greenstones along with three apatitic limestone samples from the Wadi Jilat.

Some of the (Dabba Marble) may have been mined near the Wadi Jilat in the PPNB [11, 41] located ca. 130 km distance from the Qa' Shubayqa. Other outcrops of greenstone are known from the Azraq oasis situated ca. 75 km away, and Dabba Marble sources have also been found in the Wadi al-Qattafi area in the southwestern part of the *Harra* (Gary Rollefson, personal communication). The presence of raw material, debris and debitage shows that larger fragments were brought to the settlement and manufactured there. These unworked greenstone pieces usually measure between 1–8 cm and were therefore easily portable. Analysis conducted as part of this study was too limited in scope and therefore cannot pinpoint the exact place of origin, but since there are no known sources of "greenstone" in the volcanic lava fields of the *Harra*, it is clear that the raw material was imported to the site over medium to long distances. We therefore consider the central, western and southwestern Azraq Basin as the most likely sources of most of the stone raw material found at Shubayqa 6. Sources for some of the rarer materials, such as malachite and other types of greenstone, can be found in the Wadi Faynan/

Fidan area in southern Jordan, located ca. 250 km away [60, 61]. Although the scope of the analysis carried out in the present study is limited and should be expanded, these initial results suggest that the inhabitants likely obtained some raw material from nearer sources, while receiving either raw material or finished beads from further afield.

## Production sequence

Using the sequential classification system of Wright et al. [41], it was possible to discriminate the frequency of different production stages among the disc beads in the Shubayqa 6 assemblages. A production sequence was established that consists of five main stages, which are illustrated in Fig 3. At stage 1 the roughout is an unworked primary blank, which at the next step, stage 2, is initially shaped by abrading the ends to make a more even surface. At stage 3 the roughout is perforated, which was done by drilling from both sides. Some of the studied roughouts assigned to this stage showed perforation errors caused by a misalignment of the biconical perforation. The next stage (4) represents the nearly completed bead, which besides full perforation had also been polished into a near-final bead shape. Stage 5 marks the concluding stage of production, when the surfaces of the bead had been completely smoothened.

The greatest range of production stages was recorded for beads made from greenstone, as every step is represented among this group. Disc beads made from other kinds of stone are only represented by later production stages (3–5), suggesting that they were either not made on site or only finished off there. We therefore focus on the greenstone (Fig 9). Greenstone roughouts from stage 1 are very rare (0.1%). However, initial study of the greenstone debitage and debris suggests that many of these had been initially sorted into the unworked fragments. It is therefore expected that further analysis of the debitage will increase the number of stage 1 roughouts. The fact that greenstone disc beads are well represented in every production stage together with the evidence of raw material import, suggests that their entire production cycle took place on site.

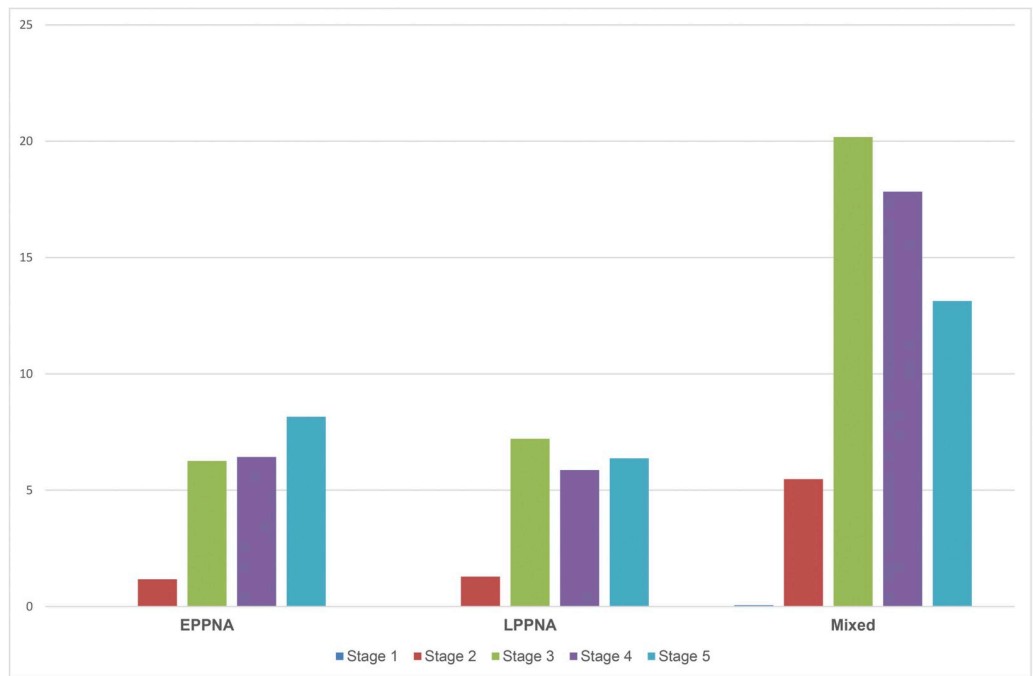

**Fig 9. Percentage distribution of production stages according to the occupational phases at Shubayqa 6.**

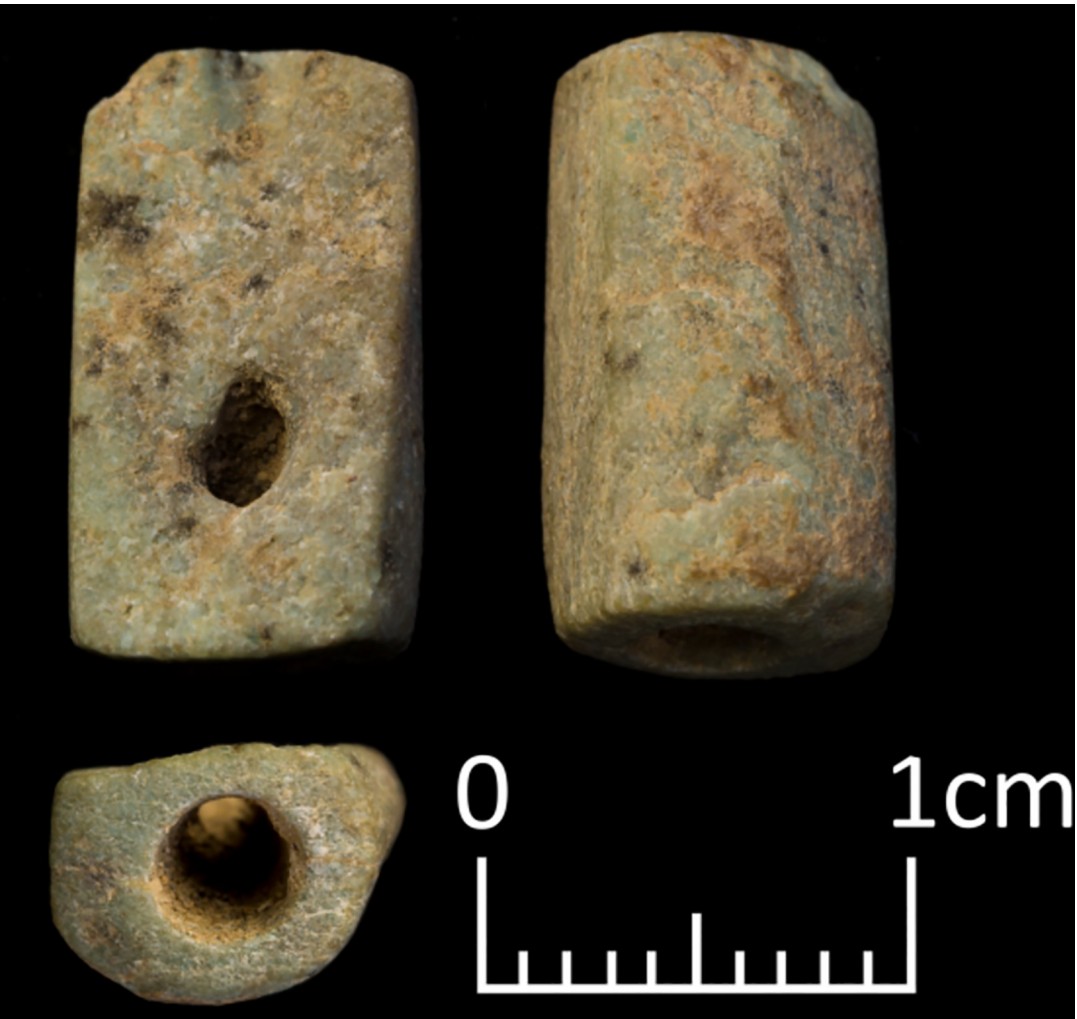

**Fig 10. Roughout that was flattened after breakage in order to still allow threading.**

There is little apparent change in this pattern between the EPPNA and LPPNA phases, and the data from mixed or post-Neolithic contexts also shows a similar pattern. There is also some evidence for the re-use of broken pieces, such as roughouts with misaligned perforations that were flattened or reshaped, so they could still be threaded (See Fig 10).

The microscopic examination disclosed more information about the production techniques. About 15% of the sampled stone beads showed abrasion marks on the profile and the surfaces of the perforation, suggesting that grinding was the main method to shape roughouts. This was done in one recurrent motion in the same direction. When it comes to the perforation stage the majority of the sampled beads appeared to have been drilled from both sides (biconically). The assessment of the perforation morphology furthermore showed that the angle of the perforation was often slanted/uneven rather than straight (see Fig 11). This resulted in occasional perforation errors, as evidenced by the finds of roughouts where the drilled holes were completely misaligned.

Wright et al. [41] observed two drilling methods in the Jilat bead assemblage: hand drilling using a borer or piercer, which was either held in the hand or hafted on a drilling stick, and

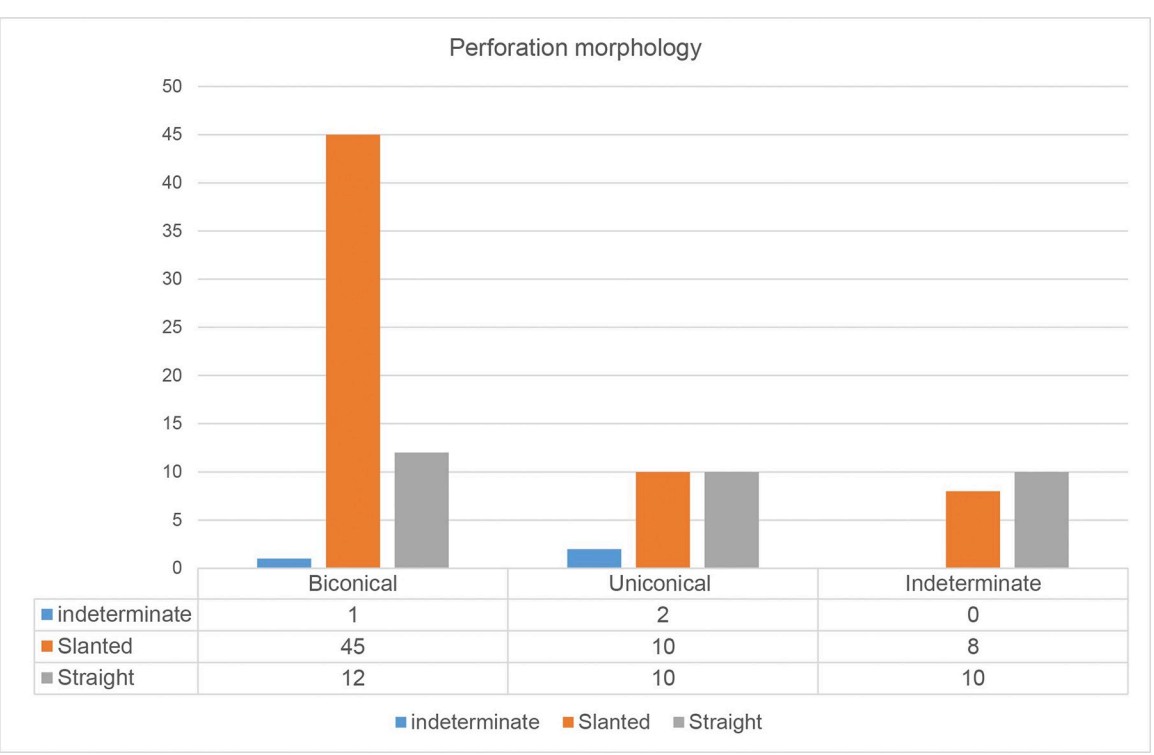

**Fig 11. Results of the microscopic analysis observing drilling technique, based on the shape of the perforation.**

mechanical drilling using a bow drill secured with a capstone. Both methods result in different perforation shapes and marks. A concentric shape and parallel-circular wear marks resulted from mechanical drilling, while slanted perforation shape and non-parallel, coarse perforation marks are related to drilling by hand (see also Bains 2012 [58]). The analysis of the perforation marks in the Shubayqa 6 assemblage was not conclusive, as most of the analysed beads had rough or indeterminate marks. Faint concentric striations, which indicate drilling by mechanical means, were primarily present in unfinished pieces on the interior sides of the perforation. Meanwhile smooth perforation marks appeared on seven out of nine finished pieces. However, these can also be caused by the final stage of polishing if the beads were strung together. In the Shubayqa 6 assemblage, three pieces exhibited traces indicating mechanical drilling, while 15 pieces had traces indicative of hand drilling. In the case of the other beads we examined, the outline of the perforation was often regular, but accompanied by slanted perforation morphology and coarse perforation marks. We interpret this combination of wear traces as drilling by hand.

When it comes to the final shaping, most of the finished beads had smoothed surfaces, in contrast to roughouts which had rougher abrasion marks, suggesting that the beads were polished during the last stage of production. The morphology of the edges, which were only observed on finished beads, also showed a clear correlation between the type of bead and the shape of the edge. The disc beads predominantly featured sharp edges, while the more uncommon types, such as barrel beads and double perforated beads all have rounded or a mixture of sharp and rounded edges. This suggests that disc beads were polished together *en masse* and other types were smoothened individually [58].

To summarise, the *chaîne opératoire* for disc greenstone beads at Shubayqa 6 can be reconstructed as follows: greenstone raw material was obtained either directly or through exchange

in areas away from the *Harra*. Nodules were brought to the settlement, where they were split by percussion. These blanks were then further reduced to roughouts by chipping and flaking. Wright et al. [41] suggest that this would have been possible by means of indirect percussion, soft-hammer percussion and pressure flaking. No retouch marks from this process were recognised on material in the Shubayqa 6 assemblage so the exact technique is not distinguishable based on the currently available data. Nevertheless, our experiments (S2 Appendix) indicate that the best way of creating flakes was by cutting grooves in the greenstone nodule and then knapping it, while it was being wedged. In some cases, saw marks were recognised on a few specimens suggesting that occasionally blanks were made by sawing regular lines halfway through the flake and then it could be almost effortlessly snapped into pieces. The experimental work further showed that this was a highly efficient way to produce square shapes, and the microscopic studies confirmed that this was the most commonly used technique by the Shubayqa 6 bead makers. Abrasion marks indicate that bead roughouts were often ground on an abrasive surface until they took on a square shape. This was likely done using a grinding tool of basalt, of which there are many at the site [48].

Once the roughout had been shaped into an approximate square the drilling of the central perforation commenced. This choice appears to have been made to streamline the production process: since drilling of the fragile disc bead roughouts carries a high risk of accidental breakage, it was done early in the process to avoid having to spend too much time on a single bead that might subsequently break during drilling. In addition, early drillings allow stringing several roughouts together in order to apply batch or *en masse* polishing. This appears to be an efficient way to reduce the overall time needed for production and highlights the experience and know-how of the Shubayqa bead makers. This process was reversed for pendants, as preforms of these indicate that drilling was the last part of the process for these types, but as these were also manufactured in larger sizes it meant that the risk of breakage during drilling was lower than for the small disc roughouts.

The majority of the sampled beads had biconical perforations, meaning that the beads were drilled from both sides using a borer or piercer, which was either held in the hand or hafted on a drilling stick. Our experiments showed that biconical drilling was the most successful strategy to avoid breakage: uniconical perforation often resulted in too much pressure which led to breakage. This work furthermore revealed that it was advantageous to have rectangular roughouts, because they made it easier to centre the perforation ensuring that holes were aligned (see figures in S2 Appendix). Preliminary analysis of the chipped stone artefacts from Shubayqa 6 shows that drilling tools make up ca. 20% of the tool assemblage from EPPNA and LPPNA contexts (see Fig 12). Although functional analysis on these tools was not performed yet, their high frequency is likely related to bead production at the site. Our own experiments provided no clear evidence that production time was significantly reduced, when abrasives (sand and water) were added during the drilling. Once perforated the roughouts were abraded one final time. Based on the uniformity in the size of the beads and the sharp edges, the disc beads were probably strung together and abraded in a cyclical motion until they had smooth surfaces and reached their final shape [58]. This stage was also successfully replicated during the experimental part of the analysis that produced beads of the same size with sharp and polished edges. When we calculate the ratio of finished to unfinished beads (roughouts), unfinished beads always outnumber finished ones slightly (see Table 2). The large amount of debitage from bead manufacture in all occupation phases at Shubayqa 6, as well as the high frequency of tools involved in bead making within the same areas underlines that bead making took place regularly.

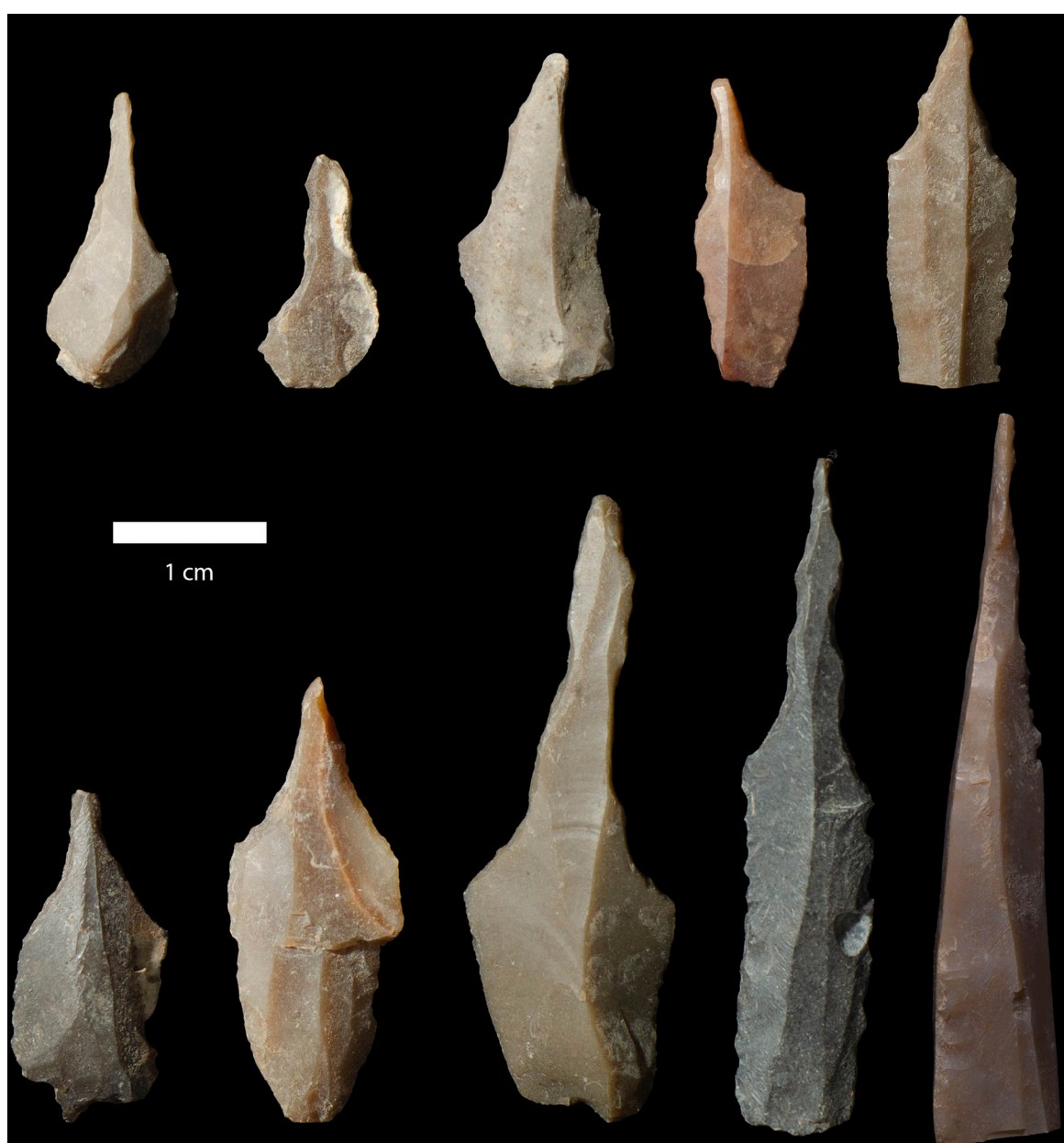

**Fig 12. Drilling tools from Shubayqa 6.**

Table 2. Ratio of finished beads in comparison to roughouts.

|  | Total amount of beads | Complete beads/versus roughouts |
|---|---|---|
| EPPNA | 533 | 1: 1.02 |
| LPPNA | 501 | 1: 1.16 |
| Mixed/Post-Neolithic | 1307 | 1: 1.58 |
| Average |  | 1: 1.255513655 |

## Discussion: Craft specialisation and the use of stone beads in the Pre-Pottery Neolithic

The identification of craft specialisation in the archaeological record has widely been regarded as a challenging task and depends on how analysts define this phenomenon. A universal definition is, however, problematic since craft specialisation is also dependent on the specific economic context and mode of production. We found Wright and Garrard's [11] criteria, reformulated after Costin [7] for the identification of artisanal activity in the archaeological record, a convenient starting point. According to this, workshops can be inferred if:

1. "Artefacts are differentially distributed among production units (households, communities, regions)

2. There is a high density of craft production debris relative to some other generally used item

3. There are high ratios of unfinished goods to finished goods." [11]

This archaeological definition eschews a more specific economic definition and instead focuses on empirical data, precisely, on what can be discerned from archaeological contexts.

What can be gathered from the presented evidence is that the stone bead assemblage from Shubayqa 6 stands out amongst other PPNA assemblages in the Levant, because of the distance between the site and the raw material sources, and the overall size of the assemblage combined with evidence for on-site production. Following the above criteria suggested by Wright and Garrard [11] Shubayqa 6 can be considered as a workshop, as these beads were arguably produced to be distributed outside the settlement. This is indicated by the high density of waste production material, and the high ratio of unfinished material to finished products. It provides evidence for an at present unique early workshop site, as there is no similar example from this period in Southwest Asia in regard to quality, quantity and the techno-economic investments. To our knowledge, the only PPNA stone bead assemblage that has also revealed thousands of beads derives from Körtik Tepe in southeast Turkey [42, 62]. However, these beads showed evidence of use and were exclusively found in burial contexts, nonetheless attesting to the importance of stone ornaments and its large-scale production in this period. In the mountains of Bal'as in central Syria, PPNA sites have revealed phosphates-based greenstones raw materials, unfinished and finished beads and pendants along with drilling tool [62, 63]. Beadmaking activities took place at these sites. The size of the assemblages of the Bal'as region is however quite small in comparison to Shubayqa 6, and the production aims seem to be focused on barrel and double perforated long beads.

The detailed technological examination of roughouts and finished beads from Shubayqa 6 suggests a highly standardised *chaîne opératoire*, in which the bead makers at Shubayqa 6 focused their attention on the usage of greenstone raw material for disc bead production. Since the nearest raw material sources are located over 60 km away it is clear that Shubayqa 6 bead makers went to considerable length to obtain this specific raw material. Stone with a reddish hue were likewise frequently used at Shubayqa 6 that possibly originate from the limestone steppe zone around the Azraq Oasis, where greenstone sources are also situated. These materials were likely obtained either directly from outcrops as part of seasonal procurement trips to the source locations in the Azraq or Wadi Jilat area or through exchange with groups resident in these areas. The presence of Dabba Marble obtained from the Wadi Jilat was confirmed by our preliminary sourcing study discussed above. The XRF study also indicated that the Wadi Jilat was not the only source of stone raw materials for bead making: at least three groups of carbonate greenstones were distinguished, suggesting significant variability and hinting at multiple sources of origin. These other, rarer types of stone were more likely

obtained as part of regional exchanges. The presence of marine shells from the Mediterranean and the Red Sea (Alarashi, ongoing analyses), as well as obsidian from Cappadocia, indicates the implication of Shubayqa 6 in a complex multi-directional network of circulation.

The efforts invested into procuring greenstone raw material from these great distances suggests that these types of raw material had a particular value or meaning that would explain why it was the preferred raw material over other types of stone that were more accessible, such as sandstone and silicified mudstone. This is further exemplified with the instances of reuse of broken roughouts, which suggest that greenstone was considered precious enough to conserve or was hard to come by. With the beginning of the 10th millennium BC, greenstone beads became popular amongst PPNA communities in the Levant (33,34). Thus, it seems that they played a significant role in PPNA societies. At Shubayqa 6 the distance to the raw material sources may have further increased their value. Traveling ~60 km in the challenging terrain of the basalt desert to obtain raw materials for bead production, either through collection at source or over shorter distances through exchange, was not necessarily a straightforward or risk-free endeavour. In many societies access to non-local raw materials helped increase the perceived value of products and access to such sources was often controlled or regulated in specific ways [7, 63]. Spielmann [63] has argued that procurement of difficult to access raw materials having particular social value is possibly an impetus towards craft specialisation, and the finds of almost whole raw material nodules at Shubayqa 6 suggests that artisans directly procured at least some of the greenstone for bead manufacture. This procurement of specific types of stones that were, due to their softness and general appearance, considered particularly suitable for the production of beads we see as indicative of specialisation at this site.

Once the raw material had been procured it was transformed into beads following a standardised manufacturing sequence that involved flaking, abrasion, perforation and polishing. All these stages were present in the archaeological record of Shubayqa 6 in the form of roughouts and were further attested by the presence of large amounts of nodules, debitage, and the high number of lithic perforators. Based on the reconstructed *chaîne opératoire* it has furthermore been possible to discern some of the technological strategies employed by the bead makers at Shubayqa 6 to accelerate and optimise the manufacturing process. This includes the observations of biconical perforations to avoid risks of breakage, the use of rectangular roughouts to more easily centre the drilling, as well as the polishing of the almost finished beads *en masse*. Thus, bead makers at Shubayqa 6 had attained a high level of skill and were able to accurately judge the qualities of the stone they worked. Combined with the scale and regularity of this workshop across the occupational sequence this provides further evidence for specialised production that was well established and also propagated (taught and learnt) through time.

Having presented the *chaîne opératoire* we will now turn our attention to the organisation of this workshop, which we consider to have consisted of part-time specialists. Archaeologists investigating the emergence of craft specialisation have traditionally leaned towards identifying it with the appearance of full-time specialists engaged in highly sophisticated craft activity, such as metalworking [64]. These activities were linked not just to very specific and detailed technological knowledge, practices, workshops and tools, but also to the production of costly prestige goods for elites. However, there is also acknowledgement of the emergence of specialist production of material culture that can be considered more mundane, such as part-time, household-based specialisation of pottery making or flint knapping [6, 10], which occurred prior to the emergence of full-time, highly specialised workers. We should not dismiss the apparent skills, knowledge and role these craft workers had, as this level of specialisation was probably much more common in most premodern societies. *Ad hoc*, temporary specialisation is for example evident in many small-scale Palaeolithic and hunting and gathering groups at certain times, and is not usually or necessarily tied to clear social differentiations or inequality [65–68].

While we have argued that there is evidence for specialised bead production at Shubayqa 6, and indeed inter-generational transmission of specialist knowledge, we cannot be certain what role or status specialist bead makers amongst the inhabitants of Shubayqa 6 had. When we look at the broader picture of the PPNA in the Levant, we can nevertheless draw on evidence from other sites to discuss what role such early craft specialists may have occupied within PPNA societies. A recurring definition of specialists are craft workers who spend all or most of their time to produce a specific type of object. However, this premise can be challenged, as it raises the question of how small-scale PPNA societies could have supported such individuals. Although some sites have produced evidence for dedicated food storage facilities [46] these are neither common nor particularly large, to enable the support of many full-time dedicated specialists. Shubayqa 6 has to date not produced any evidence for storage facilities either. How then can we reconcile the existence of specialist production with the lack of significant surplus food production? The most probable interpretation is that specialist production was carried out by those members of a group that were unable to participate in food procurement activities in a meaningful way. Alternatively, there may also have been a clearer division of tasks within a group, with one segment procuring food and the other focusing on other activities. Finally, it is also conceivable that food procurement simply did not occupy that much of the day-to-day activities and that bead making took place at times when other work was not carried out, as for example Peschaux et al. demonstrated in the Paris Basin during the Upper Magdalenian period, where bead making appeared to be seasonally organised [69]. Although we cannot resolve these issues at present, it seems likely that at least at Shubayqa 6 bead making, where it seems to have been a well-developed craft, was only ever carried out by a small and specific group of people. The key point here, however, is that the production of food surplus does not appear to be a requirement for craft specialisation.

A further question we wish to briefly reflect on relates to the products of this specialist production: the greenstone beads. What was the incentive for certain individuals to produce disc greenstone beads on a mass scale? One common reason for specialisation to emerge is that there is a demand for certain items as part of trade and exchange networks [7, 43]. Given their presence at virtually every site, greenstone beads were clearly in demand with the beginning of the PPNA, which led to the inhabitants of Shubayqa 6 to import raw material from over 60 km away. Thus, early Neolithic societies appear to have attached some value and meaning to greenstone and the beads made from them. The underrepresentation of finished versus unfinished beads at Shubayqa 6 suggests that beads were not just produced for local use or consumption, but probably for exchange. It is unclear what may have been exchanged for these beads, and it is difficult to say whether this was necessarily a purely economic exchange of goods, or whether beads were exchanged as part of other forms of social interaction (e.g. in gift exchanges, as part of matrimonial exchanges or other purposes). However, it is clear that the inhabitants of Shubayqa 6 made a high socio-economic investment into the manufacture of these beads.

The discussion on the usage and meaning of greenstone beads in early Neolithic societies usually centre on these beads being seen as artefacts that have symbolic or ritual meaning, as adornment in the negotiation and establishment of social identities and as circulated trade goods [11, 58]. According to Bar-Yosef Mayer & Porat [34], for example, greenstone beads were utilised as charm or amulets protecting the wearer from misfortune and enhancing fertility. While many of these uses and meanings are plausible, we contribute to this discussion with an economic perspective, in order to arrive at a fuller picture on the use of these beads. We suggest that greenstone beads, as well as beads made from other materials, circulated as an important means of exchange among early Neolithic societies in the Levant. We base this argument on the following observations: disc beads were common and present at most PPNA and

PPNB sites throughout the region; the raw material was not easily obtained everywhere, making the raw material rare; and, for reasons that elude us for now, greenstone and similarly colourful stones carried symbolic meaning that made them desired items. Production of greenstone disc beads as well as other beads was a time consuming endeavour that required socio-economic investment. This specialist production would have heightened the value of the final product. Beads occur at sites close to and far from raw material sources and were not produced massively at all sites. This suggests that they were a frequent medium of exchange between groups occupying different localities in the landscape.

There is ample ethnographic and archaeological evidence for the use of shell and other beads as a type of currency in societies around the world that may be used to understand the situation we seem to have in the 10th– 8th millennium BCE in southwest Asia [70, 71]. What further underlines this comparison is that in these different historical cases beads are considered to have both commercial/symbolic and functional value, as they can also be utilised as amulets or adornment. Importantly, economic anthropologists do not identify the presence of currencies with methods of payment, but foremost as a means of gift exchange, value measurement or as accounting units. Among this the exchange of standardized objects as a means to facilitate exchange is the most important, something which the disc beads could have fulfilled. Regional interaction and exchange have widely been seen as an important aspect of early Neolithic societies in the Levant and southwest Asia generally [72–77], and likely played a crucial role in the emergence of plant cultivation and animal herding practices. Such interaction networks required widely accepted forms and means of exchange and we would argue that stone beads were used in these social contexts. It can even be considered that they were introduced into a much earlier system that was based on marine shell beads, which existed since the Late Epipalaeolithic [78].

## Conclusion

The orthodox assumption about the emergence of craft specialisation in the human past has associated the appearance of craft workers with the emergence of urban, hierarchical societies, which produced sufficient surplus to enable artisans to specialise in different acts of production. Although it is also acknowledged that a certain level of specialisation is common to all societies, including mobile hunter-gatherers and pastoralists, specialisation has widely been seen as emerging from the kind of social and economic relations existing in early complex societies. In southwest Asia, incipient craft specialisation, including bead manufacture, has at times been put forward for LPPNB societies. This development has been associated with food surplus, the emergence of nuclear households, population growth, as well as emergent social inequality. The evidence from Shubayqa 6 suggests that specialised craft production has its root in a different economic and social setting that does not correspond to standard evolutionary models. Although associated with early plant cultivation, PPNA communities likely produced little significant food surplus, relying instead on a mixed subsistence strategy that involved gathering, cultivating and hunting. These communities were furthermore largely autonomous, with little evidence for social inequality or stratification.

An increasingly complex stone bead production evolved throughout the Neolithic Period in Southwest Asia, which in the southern Levant led to the widespread exchange of greenstone beads. An early production centre for these can be found at the site of Shubayqa 6, which has revealed substantial evidence of selective use of imported raw material, a standardised production sequence, and an unevenly distributed ratio of roughouts in relation to finished beads, which all hints towards nascent craft specialisation in stone bead production as early as the PPNA. It provides a case against archaeological and anthropological narratives that places the

emergence of craft specialisation in the contexts of early complex societies with food surplus production and increasing social stratification. What initiated the intensified production of greenstone beads, and why these were so widely utilised still remains open for discussion. However, we argue that the starting point for understanding this issue lies in the broader social context of their use and their possible role in the establishment and maintenance of early inter-regional exchange networks.

## Supporting information

**S1 Appendix. Shubayqa 6 XRF data.**
(XLSX)

**S2 Appendix. Greenstone bead experiment report.**
(PDF)

**S3 Appendix. Bead analysis data.**
(XLSX)

**S4 Appendix. Certificate of calibrated CRM values.**
(PDF)

**S1 Fig. The piece of green Dabba Marble from Wadi Jilat that was used for the experiment.**
(TIF)

**S2 Fig. Abrading the greenstone fragment into a rectangular preform on non-vesicular basalt.**
(TIF)

**S3 Fig. Cutting the greenstone with a flint blade.**
(TIF)

**S4 Fig. Drilling of rectangular roughout.**
(TIF)

**S5 Fig. The beads strung together with sinew.**
(TIF)

**S6 Fig. En masse polishing.**
(TIF)

**S7 Fig. A selection of the beads that were produced during this experiment.**
(TIF)

## Acknowledgments

We express our sincere gratitude to the Department of Antiquities of Jordan for giving permission to conduct fieldwork at Shubayqa 6. We are grateful to our local host community in al-Safawi, and especially Mr Ali Shreitir for help in the field. We would like to thank Alexis Pantos for the photographs reproduced in Figs 2–4 & 12 and in the bead experiment report (S2 Appendix), and Lisa Yeomans for the site plan reproduced in Fig 2 and for the photograph reproduced in Fig 10. We would also like to thank the two anonymous reviewers for their comments.

## Author Contributions

**Conceptualization:** Mette Bangsborg Thuesen, Tobias Richter.

**Data curation:** Mette Bangsborg Thuesen, Hala Alarashi, Tobias Richter.

**Formal analysis:** Mette Bangsborg Thuesen, Anthony Ruter.

**Funding acquisition:** Tobias Richter.

**Investigation:** Mette Bangsborg Thuesen, Hala Alarashi.

**Methodology:** Mette Bangsborg Thuesen, Hala Alarashi.

**Project administration:** Tobias Richter.

**Supervision:** Tobias Richter.

**Visualization:** Hala Alarashi.

**Writing – original draft:** Mette Bangsborg Thuesen.

**Writing – review & editing:** Mette Bangsborg Thuesen, Hala Alarashi, Anthony Ruter, Tobias Richter.

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
