## [Decision Letter · Decision Letter 0]

19 Apr 2023

PONE-D-23-08389Nascent craft specialization in the Pre-Pottery Neolithic A?

Skilled bead making at Shubayqa 6 (northeast Jordan)PLOS ONE

Dear Dr. Thuesen,

Thank you for submitting your manuscript to PLOS ONE. After careful consideration, we feel that it has merit but does not fully meet PLOS ONE’s publication criteria as it currently stands. Therefore, we invite you to submit a revised version of the manuscript that addresses the points raised during the review process.

We look forward to receiving your revised manuscript.

Kind regards,

Christian Reepmeyer, PhD

Academic Editor

PLOS ONE

Journal Requirements:

   "The research reported in this paper was enabled by grants from the Independent Research Fund Denmark (DFF – 4001-00068 and DFF-8018-00133B), the Danish     Institute in Damascus and the H.P. Hjerl Mindefondet for Dansk Palæstinaforskning. 

    URL:

    https://dff.dk/en

    http://damaskus.dk/

    NO " 

Additional Editor Comments:

Please have a close read through the comments and particularly address:

- Issues about the geochemical analysis of the artefacts, add CRM values to the dataset

- Issues about the re-structuring of the results/discussion section, in reference to interpretations of the data

Reviewers' comments:

Reviewer's Responses to Questions

**Comments to the Author**

1. Is the manuscript technically sound, and do the data support the conclusions?

Reviewer #1: Partly

Reviewer #2: Yes

2. Has the statistical analysis been performed appropriately and rigorously? 

Reviewer #1: N/A

Reviewer #2: N/A

3. Have the authors made all data underlying the findings in their manuscript fully available?

Reviewer #1: Yes

Reviewer #2: Yes

4. Is the manuscript presented in an intelligible fashion and written in standard English?

Reviewer #1: Yes

Reviewer #2: Yes

5. Review Comments to the Author

Reviewer #1: The paper reports on the evidence of stone bead production from PPNA contexts at a site (Shubayqa 6) in northeast Jordan. Although various types of beads were recovered, the focus is on the ‘chaîne opératoire’ of disc bead production represented by raw material nodules (mainly greenish rocks or minerals), debitage, roughouts, finished beads + knapped stone (chert) drills. The methods of analysis included macroscopic and stereomicroscopic examination of wear traces on artifacts/debitage and rock mineralogy/texture, and non-destructive chemical (pXRF) analysis of raw material samples from the site (and one external rock source). The interpretations were aided by experimental bead production. The data lead to the conclusion that bead production at Shubayqa 6 was undertaken by ‘craft specialists’ – i.e. skilled individuals who concentrated on producing beads for exchange with other households or communities, evidenced by “selective use of imported raw material, a standardized production sequence, and an unevenly distributed ratio of roughouts … to finished beads” (lines 631-632). This leads on to the suggestion that Shubayqa 6 provides the earliest evidence of craft specialization in bead production from Southwest Asia (PPNA), previously only recognized in later (PPNB) contexts.

CRITICAL OBSERVATIONS

• There is a certain amount of (unnecessary) repetition within the text. For example, lines 276-279 (“The PPM values … siliciclastic specimens”) repeat text that appears elsewhere; lines 335-343 (“The copper minerals …light elements”) repeats more or less verbatim statements elsewhere in the text (e.g. 322-329). This is unnecessary and should be removed.

• Insufficient information about the pXRF analyses is provided. Was the instrument used with/without a test stand? In all cases, were measurements made with or without the use of sample pots? What is the degree of statistical precision on the measurements reported in Table S1 – 1 standard deviation, 2SD, 3SD? Note that the abbreviation “Bal” = means “balance” (not ballast) – which is the amount of the signal the instrument is unable to attribute to an element.

• Moreover, XRF analyzers estimate chemical composition using mathematical procedures (algorithms) to “determine” the relation between the concentration of an element in the sample and the intensity of the fluorescence from that element measured by the instrument. Niton XL3 analyzers offer two mathematical procedures for measurement of geological samples – “Compton correction” (Soils mode) and “Fundamental Parameters” (Mining mode). “TestAll Geo” (note the spelling and capitalization) uses both calibration models and tries to decide which is the best for each element. Unfortunately, the output from the NDT software doesn’t report which mathematical procedure (Compton vs FP) was chosen for which element, and in my experience the “Soils” and “Mining” modes can sometimes give quite different results for the same samples. In any case, it is not advisable to trust the compositional data produced by a pXRF analyzer without performing an external calibration check – e.g. by taking measurements on Certified Reference Materials (CRMs) of known chemical composition and comparing the pXRF data with the “known” values (e.g., using linear regression) to derive correction factors (Cal factors) that can be entered into the instrument. Was this done in the case of the Shubayqa 6 samples that were analyzed?

• In several places in the text (e.g. lines 272-3), it is stated that PPM estimates for 80 elements were produced. This is not possible. At best, Niton XL3 analyzers can quantify up to 43 elements between Mg and U in the periodic table. In practice, the concentrations of some elements are so low that they are below the limits of detection (LOD) of the instrument.

• WAS BEAD PRODUCTION DONE BY SPECIALISTS?

In several places, the authors infer that specialist production is reflected in the high ratio of unfinished goods to finished goods (lines 163, 463-468, 478-481). How so? Bead making whether skilled or unskilled generates a lot of waste – in some circumstances, unskilled production may generate more waste. The ratio of unfinished to finished beads may also be affected by on-site recovery methods – nodules and roughouts are larger than finished beads, therefore more likely to be recovered in sieving. Some finished beads were smaller than the mesh size of the sieves that were used at Shubayqa 6, therefore many beads <5-mm diameter may not have been recovered. If 'dry' sieving was used on site, were sieve residues washed with water before sorting? In my experience, this is essential for recognition/recovery of small artifacts, especially those made of darker colored materials.

• From Figs 3-4 the finished beads don’t strike me as particularly skilled products. I have been involved in several bead-making workshops with University students. After a few hours training/practice, some students can produce results equivalent to what I see in the illustrations here. I don’t accept that “selective use of imported raw material, a standardized production sequence, and an unevenly distributed ratio of roughouts … to finished beads” signifies specialist rather than household production. Stronger evidence is needed!

RECOMMENDATION

That said, the paper has a good deal of merit. It is interesting, well written and presents new data from an important geographical region. But it needs some reorganization and more careful reasoning/argumentation, whist avoiding unnecessary repetition of information and observations. Present the evidence and archaeological background as objectively as possible, reserve consideration of specialist vs household for the Discussion, and draw a reasoned set of Conclusions that do not “stretch” the evidence.

Reviewer #2: The paper presents the analysis of a very large assemblage of beads and the by-products of their production at the PPNA site of Shubayqa 6. Being one of the largest known bead assemblages of beads in the PPNA of Southwest Asia, comprised of almost 2400 beads and bead roughouts as well as about 9000 fragments, it enables an in-depth study of the assemblage from a typological viewpoint as well as production sequence and raw materials. Placing this assemblage in a broader context enabled the authors to offer an interpretation of the role of craft specialization the Neolithic of the Southwest Asian societies.

The following comments should help the authors to improve their manuscript:

line 62: The period in Late Epipalaeolithic, the culture is Natufian.

line 65: stone becomes A dominant material, not THE dominant material. Shells are still an important component of ornaments.

line 114: dates of the Natufian: A date of 15 k now more accepted based on El Wad dating. See https://doi.org/10.1016/j.jhevol.2014.02.011

line 118 onwards: This wording of this part is confusing. First give the four major phases, then provide the dates (or present them in a table). Also please present them from older to younger, rather than in stratigraphic order.

lube 141 – Fig. 2: The caption for this figure should be more detailed, specifying A,B,C,D parts of the figure

lines 144, 148: the numbers of artifacts appear in the results and are not necessary in “materials and methods”.

lines 153-4, Fig. 3 : what are the five stages? above only four are mentioned (Finished beads, roughouts, debitage and nodules)

lines 155-165 do not exactly fit the “methods” section – please consider moving them to the discussion.

line 178: SI Appendix: please correct :”apetitic limestone” to apatitic (apatite).

line 195: ref 32 should probably be corrected to ref 33. This spelling mistake occurs again in Fig. 7.

lines 197-8: how are these nine stages (presumably referring ONLY to disc beads) different from the five stages mentioned in fig. 3? Please note that in Wright et al. 2008: Fig. 8, the nine stages contain "perforation error" and "perforation complete" neither of which are a stage on their own. The latter is the result of "perforation 1” and “perforation 2” combined. This is not a stage in itself. Please reconsider. Further down, on line 376 you refer again to five stages, please be more clear about this.

line 223, Table 1 and line 234, Fig. 4: 1. It would be useful to include actual numbers in addition to percentages. 2. Because there is no agreed nomenclature of bead types, it would be useful to provide a brief description of the bead types mentioned, or include a plate that shows a drawing from at least two angles (preferably three) with the names. I am not sure I know what "oval" bead or toggle bead are, and what is the difference between tubular and cylinder bead?

line 302: Calcium Carbonate is CaCO3 (3 in subscript)

line 368: Production sequence: A large portion of this section seems to belong to the discussion rather than results, because it includes many interpretations of the observations.

line 541: "prestige good for elites" is not necessarily evident in the Neolithic. Very likely there were leaders or higher-ranking individuals within the society, but the term elite, at least in archaeology, is more consistent with urban societies. In the next sentences you express a more balanced view of the place of bead making in this society, therefore I suggest to rephrase lines 539-548.

lines 564-5: Indeed, seasonality in bead making has already been shown in Palaeolithic sites, e.g., Peschaux, C, G Debout, O Bignon-Lau, and P Bodu. 2017. “Magdalenian ‘Beadwork Time’ in the Paris Basin (France): Correlation between Personal Ornaments and the Function of Archaeological Sites.” In Not Just for Show: The Archaeology of Beads, Beadwork, and Personal Ornaments, edited by D E Bar-Yosef Mayer, C Bonsall, and A M Choyke, 19–38. Not Just for Show: The Archaeology of Beads, Beadwork and Personal Ornaments. Oxford: Oxbow books.

lines 567-8: please rephrase last sentence. The link between a and b?

line 600: I disagree with the authors that there might be a connection between the Neolithic use of beads as known from the Levant and that of California and other parts of the Americas many millennia later. But the authors' are entitled to their opinion... As with the case of “elites”, I believe that the use of "shell money" is a much later invention that appears after the onset of urbanism, but I do agree that the use of beads in exchange systems was an important aspect of the Neolithic lifeway.

line 768: Proper citation for Twiss 2015:

In: Susan Pollock (ed.) Between Feasts and Daily Meals. Berlin Studies of the Ancient World 30

www.edition-topoi.de

6. PLOS authors have the option to publish the peer review history of their article (what does this mean?). If published, this will include your full peer review and any attached files.

Reviewer #1: No

Reviewer #2: No

---

## [Author Response · Author response to Decision Letter 0]

4 Jul 2023

Dear reviewers,

We thank you for your helpful comments, which have aided us in the revision and improvement of this manuscript.

Within the new manuscript you will find that we have made some changes following your suggestions. These include a clarification of the stratigraphic sequence of Shubayqa 6, a more concise description of the geo-chemical analysis, bead typology and production sequence. Some corrections regarding typos and incorrect wording throughout the text have been made, which were rightfully pointed out. We have reorganized the structure of the text in the sense that we have restricted the arguments related to craft specialization to the introduction and to the discussion, so the methods/material and the result section is presenting the evidence as objectively as possible. Although Reviewer 1 is skeptical about the argumentation behind the craft specialization, we still believe that what we consider to be skilled manufacture of disc beads at Shubayqa 6 adds important insights to the wider debate, while clarifying our perception of how craft specialization emerged in early prehistoric societies. Additionally, as we discussed, the significance of household production is traditionally devalued in the discourse aiming at identifying specialization. We do not agree that the evidence is being stretched, as we follow a clear and structured methodology to reach these conclusions.

We have highlighted all the changes within the manuscript, and all comments are addressed in detail below.

On behalf of all the authors,

Mette Bangsborg Thuesen

 

Review 1

CRITICAL OBSERVATIONS

• There is a certain amount of (unnecessary) repetition within the text. For example, lines 276-279 (“The PPM values … siliciclastic specimens”) repeat text that appears elsewhere; lines 335-343 (“The copper minerals …light elements”) repeats more or less verbatim statements elsewhere in the text (e.g. 322-329). This is unnecessary and should be removed.

Response: This redundancy was due to an editing error. Edited sections were rewritten in the google doc but the original unedited text was not deleted. Therefore, the submitted manuscript contained several uncorrected “ghost paragraphs” similar to their corrected doppelgangers. This has been corrected. 

• Insufficient information about the pXRF analyses is provided. Was the instrument used with/without a test stand? In all cases, were measurements made with or without the use of sample pots? What is the degree of statistical precision on the measurements reported in Table S1 – 1 standard deviation, 2SD, 3SD? Note that the abbreviation “Bal” = means “balance” (not ballast) – which is the amount of the signal the instrument is unable to attribute to an element.

Response: We have now updated the text and table. The instrument was used with a test stand but without sample pots. The measurements reported in the table S1 were within a three standard deviation detection limit, standard for the device. The “ballast” typo has been corrected to “balance”.

• Moreover, XRF analyzers estimate chemical composition using mathematical procedures (algorithms) to “determine” the relation between the concentration of an element in the sample and the intensity of the fluorescence from that element measured by the instrument. Niton XL3 analyzers offer two mathematical procedures for measurement of geological samples – “Compton correction” (Soils mode) and “Fundamental Parameters” (Mining mode). “TestAll Geo” (note the spelling and capitalization) uses both calibration models and tries to decide which is the best for each element. Unfortunately, the output from the NDT software doesn’t report which mathematical procedure (Compton vs FP) was chosen for which element, and in my experience the “Soils” and “Mining” modes can sometimes give quite different results for the same samples. In any case, it is not advisable to trust the compositional data produced by a pXRF analyzer without performing an external calibration check – e.g. by taking measurements on Certified Reference Materials (CRMs) of known chemical composition and comparing the pXRF data with the “known” values (e.g., using linear regression) to derive correction factors (Cal factors) that can be entered into the instrument. Was this done in the case of the Shubayqa 6 samples that were analyzed?

Response: We are confident that TestAll Geo is the most appropriate mode for this analysis. As the reviewer noted, TestAll Geo toggles between the Compton Normalisation and the Fundamental Parameters calibration. However, this is in order to minimise bias arising from differences in the ability of these procedures to measure the total metal content of the sample (Eslami et al., 2020; Lin et al., 2022). TestAll Geo is the most common mode used to analyse mixed assemblages in archeological contexts, and has been tested against other methods with satisfactory results (Williams et al., 2020).

We have several pXRF analyzers in our lab that are calibrated regularly against CRMs.

Please note that XRF results can often vary considerably from those produced by other methods like ISM-CNR. Were the objective of this study to match precisely archeological specimens to their potential geological quarries we would measure a subset of geological and archaeological samples with ISM and then calibrate the results to our pXRF analyzer. However, the more modest objective of this study as plainly stated, was only to broadly characterise the variability in the assemblage and make a preliminary comparison to a single potential source. Our methods are more than adequate to this end.

References:

Eslami, M., Wicke, D., Rajabi, N., 2020. Geochemical analyses result of prehistoric pottery from the site of Tol-e Kamin (Fars, Iran) by pXRF. STAR: Science & Technology of Archaeological Research 6, 61–71. https://doi.org/10.1080/20548923.2020.1759912

Lin, S.C., White, L.T., Jatmiko, Julianto, I.M.A., Tocheri, M.W., Sutikna, T., 2022. Characterising the stone artefact raw materials at Liang Bua, Indonesia. J Paleo Arch 6, 22. https://doi.org/10.1007/s41982-022-00133-9

Williams, R., Taylor, G., Orr, C., 2020. pXRF method development for elemental analysis of archaeological soil. Archaeometry 62, 1145–1163. https://doi.org/10.1111/arcm.12583

• In several places in the text (e.g. lines 272-3), it is stated that PPM estimates for 80 elements were produced. This is not possible. At best, Niton XL3 analyzers can quantify up to 43 elements between Mg and U in the periodic table. In practice, the concentrations of some elements are so low that they are below the limits of detection (LOD) of the instrument.

Response: This was simply an error occurring in several of the “ghost paragraphs” that should have been deleted. The actual number of elements measured was 40. This error has been corrected.

• WAS BEAD PRODUCTION DONE BY SPECIALISTS?

In several places, the authors infer that specialist production is reflected in the high ratio of unfinished goods to finished goods (lines 163, 463-468, 478-481). How so? Bead making whether skilled or unskilled generates a lot of waste – in some circumstances, unskilled production may generate more waste. The ratio of unfinished to finished beads may also be affected by on-site recovery methods – nodules and roughouts are larger than finished beads, therefore more likely to be recovered in sieving. Some finished beads were smaller than the mesh size of the sieves that were used at Shubayqa 6, therefore many beads <5-mm diameter may not have been recovered. If 'dry' sieving was used on site, were sieve residues washed with water before sorting? In my experience, this is essential for recognition/recovery of small artifacts, especially those made of darker colored materials.

Response: “In line 163 we are directly citing one of Costin’s definition of craft specialization. However, in the other sentences, which the reviewer is mentioning here, we are simply stating that the data in combination – and therefore not based on just the ratio of finished to unfinished products – suggest that Shubayqa 6 was a workshop, where bead manufacturing took place regularly given the consistent evidence throughout the different occupational phases. It is therefore this combined evidence that we suggest hints towards what might be considered nascent specialization in bead making.

Although we did have a 0.5 x 0.5 cm mesh size, the soil was often lumpy and would only break up when the material was washed in the field station. We also took bulk sediment samples for archaeobotanic analysis, where the recovery of complete beads was not much higher than those from the dry sieve residue. However, in order to clarify the recovering strategies, we have added more about the sieving strategies during the excavation of Shubayqa 6 to the introduction.”

• From Figs 3-4 the finished beads don’t strike me as particularly skilled products. I have been involved in several bead-making workshops with University students. After a few hours training/practice, some students can produce results equivalent to what I see in the illustrations here. I don’t accept that “selective use of imported raw material, a standardized production sequence, and an unevenly distributed ratio of roughouts … to finished beads” signifies specialist rather than household production. Stronger evidence is needed!

Response: "We appreciate Reviewer 1's difference of opinion with regards to the quality of the beads, but disagree with their view. The statement that the beads "don’t strike me as particularly skilled products" appears to us subjective. The argument of the ability of students to reproduce beads is not based on clear experimental protocols and other significant parameters to clarify such as the nature of the raw materials, the technicity of the elements, their sizes, the tools, etc. The comparison between university students making beads as part of their education program or in a controlled experiment setting can in no way be compared to the data we have presented, as no study is cited here that would specify the experimental setup of the bead making workshop referred to here. Some of the students may have already been skilled craftspeople and thus would have had much experience in general craft skills. Further, one cannot compare a set up in which nowadays trained individuals are given a specific goal to accomplish based on accumulated knowledge from archaeological or ethnographic finds , with PPNA hunter-gatherers who also had to find food and cook it, maintain shelters or take care of a myriad of other tasks at the same time. Finally, we do not argue that the Shubayqa 6 bead makers were full-time specialists, as is implied by the reviewers comment that this was household production. What we are proposing here is an alternative production model that does not rely on full-time dedication but is still significant and skilled enough to result in a 'nascent' specialization in stone bead crafts. We therefore stand by our argument.

RECOMMENDATION

That said, the paper has a good deal of merit. It is interesting, well written and presents new data from an important geographical region. But it needs some reorganization and more careful reasoning/argumentation, whist avoiding unnecessary repetition of information and observations. Present the evidence and archaeological background as objectively as possible, reserve consideration of specialist vs household for the Discussion, and draw a reasoned set of Conclusions that do not “stretch” the evidence.

Review 2

The following comments should help the authors to improve their manuscript:

line 62: The period in Late Epipalaeolithic, the culture is Natufian.

Response: “Yes. We have therefore rephrased this sentence slightly, although we would like to avoid using the term culture”

line 65: stone becomes A dominant material, not THE dominant material. Shells are still an important component of ornaments.

Response: “That’s a valid point and we have corrected this line according to Reviewer 2’s suggestion.” 

line 114: dates of the Natufian: A date of 15 k now more accepted based on El Wad dating. See https://doi.org/10.1016/j.jhevol.2014.02.011

Response: “We are aware of this article, but as the majority of dates from el-Wad are similar to Shubayqa and cluster around 14,5 kya BP (with only one date falling within the 15 kya BP - and then only at the 2 sigma range), we would like to keep the time frame for the Natufian as already presented in this paper.”

line 118 onwards: This wording of this part is confusing. First give the four major phases, then provide the dates (or present them in a table). Also please present them from older to younger, rather than in stratigraphic order.

Response: “The introduction text has been amended, according to Reviewer 2’s suggestions, The occupational phases are now presented chronologically.”

line 141 – Fig. 2: The caption for this figure should be more detailed, specifying A,B,C,D parts of the figure

Response: “The caption of the figure has been clarified with more detailed descriptions, paying attention to the parts of the beads shown, as suggested by the reviewer.”

lines 144, 148: the numbers of artifacts appear in the results and are not necessary in “materials and methods”.

Response: “The line has been rephrased, following Reviewer 2’s suggestion”

lines 153-4, Fig. 3: what are the five stages? above only four are mentioned (Finished beads, roughouts, debitage and nodules)

Response: “The five stages have been further clarified in the Materials and Method section. They consists of: 

1. The extraction of fragments of raw materials

2. The shaping of tabular roughout

3. The drilling of the tabulars

4. The shaping of pseudo-circular shapes of the drilled tabulars 

5. The finishing stage of the disc beads.”

lines 155-165 do not exactly fit the “methods” section – please consider moving them to the discussion.

Response: “The mentioned lines have been moved and slightly rephrased to fit the start of the discussion, as suggested”. 

line 178: SI Appendix: please correct :”apetitic limestone” to apatitic (apatite).

Response: “The typo in SI Appendix has been corrected”

line 195: ref 32 should probably be corrected to ref 33. This spelling mistake occurs again in Fig. 7.

Response: “Yes, the reference has now been corrected within the first sentence. However, this reference is not mentioned in Fig. 7, which presents the PPM values from the XRF analysis.”

lines 197-8: how are these nine stages (presumably referring ONLY to disc beads) different from the five stages mentioned in fig. 3? Please note that in Wright et al. 2008: Fig. 8, the nine stages contain "perforation error" and "perforation complete" neither of which are a stage on their own. The latter is the result of "perforation 1” and “perforation 2” combined. This is not a stage in itself. Please reconsider. Further down, on line 376 you refer again to five stages, please be more clear about this.

Response: “Yes, we realize that the phrasing may have caused some confusion and have clarified the section that describes the recognized production stages. Early on in this study the production sequence presented by Wright et al. 2008 was used as a reference model, but for the same reasons that the reviewer rightfully points out, we eventually deemed it better to define a new sequence with five stages, which is now the only one described in the manuscript”. 

line 223, Table 1 and line 234, Fig. 4: 1. It would be useful to include actual numbers in addition to percentages. 2. Because there is no agreed nomenclature of bead types, it would be useful to provide a brief description of the bead types mentioned, or include a plate that shows a drawing from at least two angles (preferably three) with the names. I am not sure I know what "oval" bead or toggle bead are, and what is the difference between tubular and cylinder bead?

Response: “We have added short descriptions for each type into the main text and the actual numbers have been added to Table 2. The figure description for Fig. 4 has been updated as well.” 

line 302: Calcium Carbonate is CaCO3 (3 in subscript)

Response: “We have corrected this typo.”

line 368: Production sequence: A large portion of this section seems to belong to the discussion rather than results, because it includes many interpretations of the observations.

Response: “We have slightly modified some sentences within the paragraphs. All sentences that are more interpretive in nature, are clearly marked as such (ie. "this suggests” and “this appears”), and we believe that these observations are necessary for understanding the reconstruction process behind the presented production sequence. We still follow clearly the stated methods and therefore their position in the chapter “results” seems justified.”

line 541: "prestige good for elites" is not necessarily evident in the Neolithic. Very likely there were leaders or higher-ranking individuals within the society, but the term elite, at least in archaeology, is more consistent with urban societies. In the next sentences you express a more balanced view of the place of bead making in this society, therefore I suggest to rephrase lines 539-548.

Response: “In this paragraph we refer to former studies on craft specialization that have traditionally focused on urbanized societies. We are therefore not arguing for the presence of elite nor elite goods in the Neolithic.”

lines 564-5: Indeed, seasonality in bead making has already been shown in Palaeolithic sites, e.g., Peschaux, C, G Debout, O Bignon-Lau, and P Bodu. 2017. “Magdalenian ‘Beadwork Time’ in the Paris Basin (France): Correlation between Personal Ornaments and the Function of Archaeological Sites.” In Not Just for Show: The Archaeology of Beads, Beadwork, and Personal Ornaments, edited by D E Bar-Yosef Mayer, C Bonsall, and A M Choyke, 19–38. Not Just for Show: The Archaeology of Beads, Beadwork and Personal Ornaments. Oxford: Oxbow books.

Response: “We thank the reviewer for this reference, which has now been added to the discussion”

lines 567-8: please rephrase last sentence. The link between a and b?

Response: “The sentence has now been rephrased” 

line 600: I disagree with the authors that there might be a connection between the Neolithic use of beads as known from the Levant and that of California and other parts of the Americas many millennia later. But the authors' are entitled to their opinion... As with the case of “elites”, I believe that the use of "shell money" is a much later invention that appears after the onset of urbanism, but I do agree that the use of beads in exchange systems was an important aspect of the Neolithic lifeway.

Response: “We understand the reviewer’s point of view, but we would like to keep these references, as we believe that they provide some new ideas into the discussion on the usage of these beads. That being said we do not want to give the impression that we are uncritically making direct analogies to the Shubayqa assemblage, and we have therefore rephrased this paragraph slightly to underline this.”

line 768: Proper citation for Twiss 2015:

In: Susan Pollock (ed.) Between Feasts and Daily Meals. Berlin Studies of the Ancient World 30

www.edition-topoi.de

Response: “This citation has been corrected, thank you.”

---

## [Decision Letter · Decision Letter 1]

1 Aug 2023

PONE-D-23-08389R1Nascent craft specialization in the Pre-Pottery Neolithic A?

Skilled bead making at Shubayqa 6 (northeast Jordan)PLOS ONE

Dear Dr. Thuesen,

Thank you for submitting your manuscript to PLOS ONE. After careful consideration, we feel that it has merit but does not fully meet PLOS ONE’s publication criteria as it currently stands. Therefore, we invite you to submit a revised version of the manuscript that addresses the points raised during the review process. Please submit your revised manuscript by Sep 15 2023 11:59PM. If you will need more time than this to complete your revisions, please reply to this message or contact the journal office at plosone@plos.org. Please include the following items when submitting your revised manuscript:A rebuttal letter that responds to each point raised by the academic editor and reviewer(s). You should upload this letter as a separate file labeled 'Response to Reviewers'.A marked-up copy of your manuscript that highlights changes made to the original version. You should upload this as a separate file labeled 'Revised Manuscript with Track Changes'.An unmarked version of your revised paper without tracked changes. You should upload this as a separate file labeled 'Manuscript'.If applicable, we recommend that you deposit your laboratory protocols in protocols.io to enhance the reproducibility of your results. Protocols.io assigns your protocol its own identifier (DOI) so that it can be cited independently in the future. For instructions see: https://journals.plos.org/plosone/s/submission-guidelines#loc-laboratory-protocols. Additionally, PLOS ONE offers an option for publishing peer-reviewed Lab Protocol articles, which describe protocols hosted on protocols.io. Read more information on sharing protocols at https://plos.org/protocols?utm_medium=editorial-email&utm_source=authorletters&utm_campaign=protocols.

We look forward to receiving your revised manuscript.

Kind regards,

Christian Reepmeyer, PhD

Academic Editor

PLOS ONE

Journal Requirements:

**Additional Editor Comments:**

Please particularly review comments from Reviewer 1 concerning the addition of CRM values into the XRF data table. The addition of calibrated XRF measurements of one or two standards against the certified values would be sufficient.

Reviewers' comments:

Reviewer's Responses to Questions

**Comments to the Author**

1. If the authors have adequately addressed your comments raised in a previous round of review and you feel that this manuscript is now acceptable for publication, you may indicate that here to bypass the “Comments to the Author” section, enter your conflict of interest statement in the “Confidential to Editor” section, and submit your "Accept" recommendation.

Reviewer #1: (No Response)

Reviewer #2: All comments have been addressed

2. Is the manuscript technically sound, and do the data support the conclusions?

Reviewer #1: Partly

Reviewer #2: Yes

3. Has the statistical analysis been performed appropriately and rigorously? 

Reviewer #1: Yes

Reviewer #2: Yes

4. Have the authors made all data underlying the findings in their manuscript fully available?

Reviewer #1: Yes

Reviewer #2: Yes

5. Is the manuscript presented in an intelligible fashion and written in standard English?

Reviewer #1: Yes

Reviewer #2: Yes

6. Review Comments to the Author

Reviewer #1: The authors have made a number of changes in response referees’ comments, which have improved the structure of the paper. However, I am not entirely satisfied by their responses to two issues that I raised:

1) They regard as “subjective” my comment that the level of skill reflected in the majority of finished beads from Shubayqa 6 is not particularly high. “Skilled” simply implies knowledge and ability to perform a task well. My assessment of the degree of skill involved is based partly on the authors’ illustrations and partly on my own archaeological and experimental observations. The authors hold a different opinion, but they are being equally “subjective”.

2) The authors may have misunderstood my comments about the need for calibration of a pXRF analyzer against CRMs. It is standard practice in chemical fingerprinting using pXRF to establish an external calibration by analyzing, say, 25–30 CRMs (e.g., powdered rock samples pressed into pellets) then comparing the results from the analyzer (for each element of interest) with the known values for each CRM, using linear regression analysis – from which you can derive a calibration factor to (hopefully) improve the accuracy of the results. The procedure is fairly straightforward – see, e.g., Simandl GJ et al. (2014) ‘Portable X-ray fluorescence in the assessment of rare earth element-enriched sedimentary phosphate deposits’. Geochemistry: Exploration, Environment, Analysis 14, 161–169. It is considered bad practice to publish the results obtained directly from the analyzer without first performing a regression-based correction of those results (see Speakman RJ & Shackley MS (2013) ‘Silo science and portable XRF in archaeology: a response to Frahm’. Journal of Archaeological Science 40, 1435–1443). What suggests to me that the authors have not used a regression-based correction of their pXRF data is that they have operated their Niton analyzer in TestAll Geo mode, which is not readily compatible with a regression-based calibration approach.

N.B. I omitted to mention in my original review that in the Methods section (pXRF) it should be stated whether the XL3 analyzer was operated with the spot size set to 8mm or 3mm diameter.

Overall, while this revised version of the text is an improvement on the original, there is a little way to go especially with the reporting of the pXRF procedures and data. There are also some typographical/grammatical alterations needed (see attached WORD file).

Reviewer #2: The comments have been addressed in an approriate way and in my opinion the paper is ready for publication. My only last minor comment is that Timna marked on the map (Fig. 1) should be placed much further to the South.

7. PLOS authors have the option to publish the peer review history of their article (what does this mean?). If published, this will include your full peer review and any attached files.

Reviewer #1: No

Reviewer #2: No

---

## [Author Response · Author response to Decision Letter 1]

27 Aug 2023

We thank the reviewers for taking the time to consider the resubmitted version of our manuscript. Herewith our responses to the comments they have raised. 

Reviewer 1)

We have presented data that suggests to us – having examined the entire assemblage directly – that the bead makers at Shubayqa 6 had attained a considerable level of skill in producing different types of beads, stone disc beads being the dominant type. Since there is no universally accepted measure for what being skilled constitutes, we certainly accept that there is always a level of subjectivity engrained in interpretation. Nevertheless, since some of us have analysed several assemblages from the same region and time period, we consider this material to stand out. We suggest a nascent specialization in stone bead craft at Shubayqa 6 based on the following points: 

- The assemblage is comprised of a high amount of both finished and unfinished beads in addition to a considerable quantity of raw materials brought to the settlement. We have recently revised the count of approximately 30% of the raw materials used to craft the disc beads at the site. These amounts to a total of 13,695 fragments of varying sizes (we have updated this information in the manuscript). Based on this calculation, we can project that for the remaining 70% of raw materials, the number of fragments could potentially double, if not triple, exceeding 40,000 fragments. This would likely translate to over 7 kilograms of raw materials (while the weight estimation is still in progress, an estimate of 1,700 grams was determined for around 9,000 fragments). Thus far, such a quantity was never recorded elsewhere for this period. Therefore, it is quite reasonable to suggest an important demand for stone disc beads. In this sense, if bead makers of Shubayqa 6 lacked the expertise and the skills to meet this demand, then how should we interpret these numbers? 

- We partially concur that from a technological perspective, crafting a disc bead can be within the means of individuals with limited artisanal skills, although this would largely hinge on factors like the nature of the raw material and the quality/adequacy of tools. Nonetheless, our interpretation takes into account the entirety of the assemblage and its archaeological and chrono-cultural contexts. The assemblage has specific characteristics such as a preferred bead type, preferred color, standardisation of the size, and a normalized method of manufacture. It was found alongside a significant quantity of piercing tools and ground stones across all the occupational PPNA levels, thus indicating a constant/continuous production at the settlement. These aspects are persuasive for us to posit that there existed an organizational pattern for crafting stone beads at the settlement. More notably, there seems to have been an intergenerational transmission of know-how. We believe that such a transmission would necessarily involve skilled artisans, probably of distinct degrees of expertise. 

Hence, we feel justified in proposing the emergence of specialization in stone bead crafting at Shubayqa 6 involving skilled or highly skilled individuals, while acknowledging that others may disagree. Following this discussion, we have, however, decided to remove “skilled” from the title of the paper as suggested and also fixed the typographical/grammatical mistakes correctly pointed out by the reviewer.

Concerning the XRF-analysis we feel that the reviewer may have misunderstood the goal and purpose of the analysis, which we carried out. The goal of the pXRF-analysis was to characterize the range of different raw materials and – in those cases where beads were very small – to actually determine whether the material was a form of rock, shell or something else. In other words, we wanted to have some idea of whether the greenstone was:

1) comprised mostly or completely of apatitic limestone (which it clearly was not since copper ores were present) and…

2) to gauge the homogeneity of the greenstone that was comprised of apatitic limestone. 

For these objectives the TestAll Geo mode is adequate, and we have consulted on this with other in-house experts in XRF-analysis. 

Only three, Wadi Jilat specimens were included in the final comparison (far too few to assess the variance within that quarry) simply to see, where they fell relative to the Shubayqa samples. This was done in order to preliminarily access if this site could have been a source for any of these samples. We did not claim that any of the Shubayqa samples originated there, only that it was an unlikely source for most of them, while it still could not be ruled out as a potential source for some of the Shubayqa specimens. This is a very conservative interpretation, consistent with the methods used. 

It is not possible at this moment for us to repeat the XRF component of the study, as the material is no longer available. If the reviewer/editors are unsatisfied with the state of the XRF component of the study, then we suggest taking that component of the paper out of the manuscript altogether. 

Reviewer 2) 

We thank Reviewer 2 for spotting the misplacement of Timna. We have adjusted the position on the map and are submitting a revised Figure 1.

---

## [Editor Report · Decision Letter 2]

5 Sep 2023

PONE-D-23-08389R2Nascent craft specialization in the Pre-Pottery Neolithic A?

Bead making at Shubayqa 6 (northeast Jordan)PLOS ONE

Dear Dr. Thuesen,

Thank you for submitting your manuscript to PLOS ONE. After careful consideration, we feel that it has merit but does not fully meet PLOS ONE’s publication criteria as it currently stands. Therefore, we invite you to submit a revised version of the manuscript that addresses the points raised during the review process.

We look forward to receiving your revised manuscript.

Kind regards,

Christian Reepmeyer, PhD

Academic Editor

PLOS ONE

Journal Requirements:

**Additional Editor Comments:**

We accept all changes to the manuscript. However, without providing the values of one or two certified standards, it is impossible for us to assess the validity of the geochemical results. We accept that the calibration of the data is done with the manufacturer provided calibration curves, we also accept that the TestAll Geo mode is adequate for the research you have been conducting.

Providing CRM (certified reference material) values is standard praxis in publishing geochemical data. Your in-house XRF experts should have this data readily available as it is standard procedure to maintain and assess long-term validity and reproducibility of the geochemical data, and the equipment itself. Please provide this data and we can proceed with the publication of this interesting paper.

---

## [Author Response · Author response to Decision Letter 2]

8 Sep 2023

We accept all changes to the manuscript. However, without providing the values of one or two certified standards, it is impossible for us to assess the validity of the geochemical results. We accept that the calibration of the data is done with the manufacturer provided calibration curves, we also accept that the TestAll Geo mode is adequate for the research you have been conducting.

Providing CRM (certified reference material) values is standard praxis in publishing geochemical data. Your in-house XRF experts should have this data readily available as it is standard procedure to maintain and assess long-term validity and reproducibility of the geochemical data, and the equipment itself. Please provide this data and we can proceed with the publication of this interesting paper.

Response: 

We have added an additional document to the Appendix (S4), which provides a certificate of calibration from May 2017 for the NITON XL3 GOLDD+ pXRF energy dispersion x-ray analyzer that was used for the geochemical analysis. The XRF measurements were made in the months of September and October of 2017, and subsequent calibrations have not registered deviations larger than those in this report, all of which were well within acceptable parameters.

---

## [Editor Report · Decision Letter 3]

13 Sep 2023

PONE-D-23-08389R3Nascent craft specialization in the Pre-Pottery Neolithic A?

Bead making at Shubayqa 6 (northeast Jordan)PLOS ONE

Dear Dr. Thuesen,

Thank you for submitting your manuscript to PLOS ONE. After careful consideration, we feel that it has merit but does not fully meet PLOS ONE’s publication criteria as it currently stands. Therefore, we invite you to submit a revised version of the manuscript that addresses the points raised during the review process.

Thank you very much for providing the manufacture calibration, however, as we said in the last response "We accept that the calibration of the data is done with the manufacturer provided calibration curves, we also accept that the TestAll Geo mode is adequate for the research you have been conducting." This information has already been accepted as accurate.If you can't provide the data for your independent measurements of a standard, can you please reference in the text which standard you have been using to validate your results? I refer you to the recent PlosOne article Alapont et al. 2023, The casts of Pompeii, https://doi.org/10.1371/journal.pone.0289378. please check the 'methods' section, for a description of this procedure.

We look forward to receiving your revised manuscript.

Kind regards,

Christian Reepmeyer, PhD

Academic Editor

PLOS ONE
---

## [Author Response · Author response to Decision Letter 3]

26 Sep 2023

Within the new manuscript we have added the following description of the standards that were used to validate the results of the chemical analysis (lines 205-208): “Certified standard reference samples including CCRMP TILL-4PP (180–646) and NIST 2709a (180–649) and a single piece of obsidian from the Fantale stratovolcano in Ethiopia, previously analyzed with ICP-MS were used to gauge the accuracy of the elements measured in this analysis. None of the measurements had greater than a 5% error compared to these standards”. 

For the reference list we have removed the following citation: 

Campbell S, Healey E. Manchester Obsidian Laboratory, Lab Report 130. University of Manchester; 2020

This refers to an unpublished laboratory report, which is not possible for us to append to this article. 

Furthermore, as we quote Prof. Gary Rollefson for personal communication regarding the presence of dabba marble sources in the Wadi al Qattafi area (line 334), we have uploaded a supportive letter from Prof. Rollefson confirming these statements.

---

## [Editor Report · Decision Letter 4]

3 Oct 2023

Nascent craft specialization in the Pre-Pottery Neolithic A?

Bead making at Shubayqa 6 (northeast Jordan)

PONE-D-23-08389R4

Dear Dr. Thuesen,

We’re pleased to inform you that your manuscript has been judged scientifically suitable for publication and will be formally accepted for publication once it meets all outstanding technical requirements.

Kind regards,

Christian Reepmeyer, PhD

Academic Editor

PLOS ONE
---

## [Editor Report · Acceptance letter]

23 Oct 2023

PONE-D-23-08389R4 

Nascent craft specialization in the Pre-Pottery Neolithic A?
Bead making at Shubayqa 6 (northeast Jordan) 

Dear Dr. Thuesen:

I'm pleased to inform you that your manuscript has been deemed suitable for publication in PLOS ONE. Congratulations! Your manuscript is now with our production department. 

Kind regards, 

on behalf of

Dr. Christian Reepmeyer 

Academic Editor

PLOS ONE